# Therapeutic Targeting of Stat3 Using Lipopolyplex Nanoparticle-Formulated siRNA in a Syngeneic Orthotopic Mouse Glioma Model

**DOI:** 10.3390/cancers11030333

**Published:** 2019-03-08

**Authors:** Benedikt Linder, Ulrike Weirauch, Alexander Ewe, Anja Uhmann, Volker Seifert, Michel Mittelbronn, Patrick N. Harter, Achim Aigner, Donat Kögel

**Affiliations:** 1Experimental Neurosurgery, Department of Neurosurgery, Neuroscience Center, Goethe University Hospital, 60528 Frankfurt am Main, Germany; koegel@em.uni-frankfurt.de; 2Rudolf-Boehm-Institute for Pharmacology and Toxicology, Clinical Pharmacology, University of Leipzig, 04107 Leipzig, Germany; Ulrike.Weirauch@medizin.uni-leipzig.de (U.W.); Alexander.Ewe@medizin.uni-leipzig.de (A.E.); Achim.Aigner@medizin.uni-leipzig.de (A.A.); 3Institute of Human Genetics, Tumor Genetics Group, University of Göttingen, 37073 Göttingen, Germany; auhmann@gwdg.de; 4Department of Neurosurgery, Goethe University Hospital, 60528 Frankfurt am Main, Germany; V.seifert@em.uni-frankfurt.de; 5Institute of Neurology (Edinger-Institute), University Hospital Frankfurt, Goethe University, 60528 Frankfurt am Main, Germany; michel.mittelbronn@lns.etat.lu (M.M.); patrick.harter@kgu.de (P.N.H.); 6Luxembourg Centre for Systems Biomedicine (LCSB), University of Luxembourg, 4362 Esch-sur-Alzette, Luxembourg; 7Laboratoire national de santé (LNS), 3555 Dudelange, Luxembourg; 8Luxembourg Centre of Neuropathology (LCNP), 3555 Dudelange, Luxembourg; 9NORLUX Neuro-Oncology Laboratory, Luxembourg Institute of Health (LIH), 1526 Luxembourg, Luxembourg; 10German Cancer Consortium DKTK Partner site Frankfurt/Main, Frankfurt am Main, Germany and German Cancer Research Center DKFZ, 69120 Heidelberg, Germany

**Keywords:** STAT3, siRNA/RNAi, polyethylenimine, PEI, lipopolyplex, siRNA delivery, glioma, glioblastoma

## Abstract

Glioblastoma (GBM), WHO grade IV, is the most aggressive primary brain tumor in adults. The median survival time using standard therapy is only 12–15 months with a 5-year survival rate of around 5%. Thus, new and effective treatment modalities are of significant importance. Signal transducer and activator of transcription 3 (Stat3) is a key signaling protein driving major hallmarks of cancer and represents a promising target for the development of targeted glioblastoma therapies. Here we present data showing that the therapeutic application of siRNAs, formulated in nanoscale lipopolyplexes (LPP) based on polyethylenimine (PEI) and the phospholipid 1,2-dipalmitoyl-sn-glycero-3-phosphocholine (DPPC), represents a promising new approach to target Stat3 in glioma. We demonstrate that the LPP-mediated delivery of siRNA mediates efficient knockdown of Stat3, suppresses Stat3 activity and limits cell growth in murine (Tu2449) and human (U87, Mz18) glioma cells in vitro. In a therapeutic setting, intracranial application of the siRNA-containing LPP leads to knockdown of STAT3 target gene expression, decreased tumor growth and significantly prolonged survival in Tu2449 glioma-bearing mice compared to negative control-treated animals. This is a proof-of-concept study introducing PEI-based lipopolyplexes as an efficient strategy for therapeutically targeting oncoproteins with otherwise limited druggability.

## 1. Introduction

Glioblastoma (grade IV glioma, WHO) is the most aggressive primary brain tumor in adults, with a median survival of ~15 months following standard radiochemotherapy with temozolomide (TMZ) and a 5-year survival rate below 5% [1]. Due to diffusely infiltrating tumor cells that cannot be surgically resected, most tumors will quickly recur and afflict the patients with even more aggressive disease [2]. GBM are further subdivided into distinct subtypes based on their genetic profile. Different groups reported different subtypes, but it is generally agreed that a proneural subtype with the best prognosis and a mesenchymal subtype with the worst prognosis exist [3,4]. Another subtype is the classical one [4], which is sometimes subdivided into a proneural and a neural subtype [3]. One key molecule that is frequently highly expressed and overactivated in GBM and associated with the most aggressive and treatment-resistant mesenchymal subtype [5,6] is the oncogenic transcription factor signal transducer and activator of transcription 3 (Stat3) that acts as a signaling hub promoting most hallmarks of cancer, including proliferation, cell survival, angiogenesis and immune evasion (reviewed in [7,8]). STAT3 activation is facilitated by phosphorylation of multiple upstream kinases [9]. Stat3 can be phosphorylated on two different sites (reviewed in [10]), Tyrosine705 (p^Y705^Stat3) and Serine727 (p^S727^Stat3), which can exert different functions in different cell types. In glioma it was shown that S727-phosphorylation is dependent on Y705-phoshorylation, that is necessary for maximal activation of Stat3 [11]. Activated Stat3 dimerizes and induces gene expression of a variety of genes, many of which are known to be important for hallmarks of cancers like migration/invasion (Mmp2, Mmp9; [12,13]) or EMT-like features and immune evasion/suppression (e.g., Snai1; [14,15]). Stat3 is activated in many other human cancers (reviewed in [16]) in addition to gliomas. In particular Stat3 expression increases with tumor grade in these tumors (Appendix A and reviewed in [9,17]), and is negatively correlated with patient survival (reviewed in [9,17]). Additionally, Stat3 is known to be necessary for activation of microglia [18], the macrophage-like population of immune cells residing in the brain. In glioma, these cells usually acquire a pro-tumorigenic phenotype that promotes glioma cell migration and invasion [18].

Consequently, we and others previously demonstrated that targeting Stat3 by upstream signaling inhibitors of the Stat3 pathway decreases glioma cell proliferation and migration in vitro and prolongs overall survival of tumor-bearing mice in vivo [19,20,21,22,23,24].

One strategy to target aberrantly activated and/or overexpressed oncoproteins in tumor therapy is to ablate their expression by RNA interference (RNAi). This knockdown strategy relies on small interfering RNA (siRNA) that, due to high sequence specificity, generally exhibits fewer off-target effects in comparison to the application of pharmacological agents (reviewed in [25,26]). The second major advantage of RNAi is that it allows to selectively interfere with the activity of otherwise un-druggable or hard-to-drug targets like Stat3 and other oncogenic transcription factors. Using a pre-transplantational genetic knockdown approach, we could previously demonstrate that ablation of Stat3 function with lentiviral shRNA limits tumor growth of Tu2449 gliomas in an orthotopic, syngeneic mouse model [21], providing support for the concept of therapeutic intervention with Stat3 activity in vivo. Therapeutic intervention in already established glioma, however, requires the development of siRNA formulations for their direct application. We have previously established siRNA complexation with nanoscale polyethlyenimine (PEI)-based complexes for therapeutic siRNA delivery in vivo. This confers protection, cellular delivery and intracellular release of formulated small RNA molecules (siRNAs, miRNAs) in vitro and in vivo, and thus allows for their therapeutic application ([27,28,29,30]; see [31] for review). More recently, this was extended towards combinations with liposomes, leading to lipopolyplexes (LPP) that combine properties of both components [32].

For gene delivery, LPP comprising phospholipids had been previously shown to display strongly reduced surface charges, enhanced transfection efficiencies, decreased cytotoxicity and high colloidal stability as compared to their unmodified complex counterparts [33,34]. Likewise, we were able to demonstrate for siRNA delivery that DPPC-based LPP without co-lipids displayed very good transfection efficiency and further enhanced biocompatibility in cell culture. Intracellular siRNA release was not impaired by the liposomal component. Due to strongly reduced surface charges, storage stabilities under various conditions were markedly increased by protecting the LPP from aggregation [35]. Importantly, these favorable properties also translated into therapeutic in vivo efficacy upon systemic injection into tumor xenograft-bearing mice [32], thus providing the basis for now switching to another model and another mode of administration.

Here we investigated the therapeutic potential of intracranially applied LPP containing Stat3-specific siRNA in an orthotopic Tu2449 glioma model. We demonstrate that specific targeting of Stat3 using LPP significantly reduces tumor growth and improves overall survival. This proof-of-concept study thus supports the notion that highly therapy-resistant cancers such as malignant gliomas and hard-to-drug genes like Stat3 can be efficiently targeted using siRNA in vivo.

## 2. Results

### 2.1. LPP Nanoparticles Provide a Stable and Efficient siRNA Formulation

As shown previously, the complexation of siRNAs with the ~4–10 kDa low molecular weight polyethylenimine PEI F25-LMW [36] leads to the formation of positively charged nanoscale complexes (‘polyplexes’) in a size range of ~100–350 nm, dependent on the buffer conditions used for complexation [27]. This was confirmed here by NanoSight measurements, yielding PEI/siRNA complexes of ~130 nm diameter with a zeta potential of 27 mV (Figure 1a,c). The combination of these polyplexes with DPPC liposomes of about the same size (Figure 1a,b) resulted in the formation of slightly larger lipopolyplexes (Figure 1a,b,d; [37,38]). The liposomal contribution also affected the zeta potential of the LPP, which was almost neutral as opposed to the polyplexes (Figure 1a). Size determinations by NanoSight measurements were confirmed by dynamic light scattering (DLS) which, however, also indicated the tendency of PEI/siRNA complexes to aggregate, leading to larger particle sizes (Figure 1a, Appendix A). Notably, this was inhibited by lipopolyplex (LPP) formation (Figure 1a; note PEI/siRNA complex size in DLS measurements). The colloidal stability of LPP provides an advantage over polyplexes and prompted us to prefer LPP for intracranial siRNA delivery. In an LDH release assay in Tu2449 cells, the LPP showed excellent biocompatibility in vitro (Figure 1e). The absence of appreciable cell damage upon transfection suggests the applicability of LPP also in a sensitive environment in vivo (CNS), and similar results in the case of the parent polyplexes also indicates that even an LPP decomposition with possible polyplex release would not lead to toxic effects. Since highly positive surface charges of nanoparticles may lead to enhanced cellular uptake, we tested next whether the considerably reduced zeta potentials of LPP would negatively affect biological efficacies. Notably, upon transfection of stable EGFP (Figure 1f) or luciferase (Luc3) reporter cell lines (Figure 1g) with siRNAs formulated in polyplexes or in LPP, no differences were observed. More specifically, the comparison of cells transfected with negative control siRNA (siLuc2) vs specific siRNA (siEGFP or siLuc3, respectively) revealed a profound 50–65% knockdown after 72 h, indicating biological efficacy of both, polyplexes and LPP. For the reasons stated above, LPP were selected for further studies.

### 2.2. Targeting STAT3 Reduces Cancer Cell Proliferation In Vitro

To test the therapeutic potential of LPP-formulated siRNA in vitro and in vivo, we selected Stat3 as a target because it is frequently overactivated in glioma and also hard-to-drug. Consistent with a number of previous reports, analysis of the TCGA data-set indicated that high expression of STAT3 is associated with an especially poor survival of GBM patients (Figure 2a). To validate these clinical observations in vitro, we knocked down STAT3 in two human GBM cell lines, U87 (Figure 2b) and Mz18 (Figure 2c). Using two different siRNAs we could significantly reduce *STAT3* mRNA expression in both cell lines, with siSTAT3-2 being more effective than siSTAT3-1. Consistently, STAT3 suppression was also achieved on the protein level in both cell lines (Figure 2d). Notably, we frequently observed a second band below the STAT3 signal in U87, but since both siRNAs target all three protein coding sequences of STAT3 (NM_213662.1, NM_003150.3 and NM_139276.2) this is likely an unspecific signal. To assess the antitumor effects of STAT3 depletion, we evaluated the growth kinetics of U87 (Figure 2e) and Mz18 cells (Figure 2f) after siSTAT3 treatment. Both cell lines showed significantly reduced proliferation 192 h after siSTAT3-treatment, with siSTAT3-2 again being more effective than siSTAT3-1. Of note, U87 cells were more sensitive to STAT3 depletion than Mz18 cells, indicating that this line may be particularly addicted to STAT3 activity, in line with findings described earlier [39]. Mz18 cells also express STAT3 and we could previously show that this line exhibits moderate levels of tyrosine-phosphorylated STAT3, which could be inhibited by upstream JAK2-inhibition [22]. We also tested the murine GBM cell line Tu2449, which we previously had used for in vivo experiments with pre-transplantational depletion of Stat3 with shRNA [21]. First, we sought out to test if siRNA-mediated Stat3-knockdown also inhibits proliferation and indeed we observed that siRNA delivery using conventional in vitro reagents like INTERFERin^TM^ also achieved a reduction in proliferation (Figure 2g). Next, we applied siRNA complexed as polyplexes, in order to verify that the delivery method does not affect knockdown efficiency. Accordingly, LPP mediated siStat3 delivery strongly inhibited proliferation (Figure 2h) and was able to efficiently reduce Stat3 and phospho-Stat3 protein levels (Figure 2i), whereas polyplexes without liposomal content were accompanied by increased nonspecific toxicities although a knockdown could also be achieved (data not shown). Thus, in these experiments LPP were found to be superior over polyplexes.

Cell cycle analysis of Tu2449 cells showed a significant increase in G1 phase and concomitant decrease in G2 phase upon siStat3 transfection, suggesting that the observed antiproliferative effect is at least in part due to a G1 arrest upon Stat3 knockdown (Figure 3a). Decreased cell cycle progression was also confirmed in the human cell lines U87 and Mz18 (Appendix A). To further verify the dependency of Tu2449 cells on Stat3 in a more complex cell culture system, we generated Tu2449 tumor spheroids, which resemble an in vivo situation more closely with regard to gradient access to oxygen, nutrients, as well as therapeutics. siRNA-mediated knockdown of Stat3 lead to distinctly smaller spheroids than control treatment (Figure 3b,c), also demonstrating that LPP are efficient in transfecting cells in spheroids.

In conclusion, we could successfully establish siRNA-mediated STAT3/Stat3 depletion and suppression of tumor cell proliferation by LPP siSTAT3 in vitro and verify Tu2449 as suitable cell line to explore a Stat3 targeting therapy.

### 2.3. LPP siStat3 Prolongs Overall Survival and Reduces Tumor Size in Glioma-Bearing Mice

Based on these findings, we pushed our system further towards therapeutic intervention, using a complex and pathophysiological relevant system. To this end, we employed our syngenic, orthotopic transplantation model [41] to assess LPP applicability and efficacy in vivo. The treatment scheme is outlined in Figure 4a. Briefly, after implantation of 10,000 Tu2449 cells into the striatum of 42 mice, the tumors were allowed to grow for one week. Hereafter, the mice were randomly divided in two groups of 21 animals each and the intracranial (i.e., intratumoral) treatment with LPPs was performed every third day. After three weeks of tumor growth, 8 mice were sacrificed for histological and molecular analyses. 1 tumor in the LPP siCtrl and LPP siStat3 cohort planned for histological assessment did not grow and were removed from the analysis. The remaining mice were monitored until they succumbed to the disease. Of those, 2 and 4 tumors of the LPP siCtrl and LPP siStat3 cohort did not grow well, respectively, and these animals were also removed from the analysis. The survival analysis (Figure 4b) showed that after LPP siStat3 none of the mice died during the treatment period, whereas six mice treated with LPP siCtrl had to be euthanized. The median survival was significantly improved after treatment with LPP siStat3 in comparison to LPP siCtrl from 26 to 33 days, respectively. Histological analyses (Figure 4c,d) confirmed that the tumors of mice treated with LPP siStat3 were significantly smaller compared to LPP siCtrl treated mice.

### 2.4. LPP siStat3 Slightly Reduces Stat3 Activation, But Does Not Affect Gross Tumor Proliferation Rates In Vivo

Next, we analyzed LPP siStat3 effects on the cellular and molecular level in vivo. When performing analyses of randomly chosen tumor areas (excluding the tumor border and necrotic areas), we found phospho-Stat3 (Tyr705) slightly reduced (Figure 5b). This, however, was not accompanied by alterations in proliferation rates, as determined by Ki67 staining for proliferating cells (Figure 5a). Decreased phospho-Stat3 levels were also confirmed in western blot analyses of whole tumor lysates (Figure 5c, upper panel), while Stat3 protein levels remained stable. Since Stat3 was proposed to modulate the host immune response including activation of microglia [18], we also analyzed the presence of T-cells and microglia in the tumor. Staining against CD4 and CD8a for T-cells and Iba1 for microglia showed a slight reduction of CD4-positive cells after LPP siStat3 treatment (Figure 6a) while no major differences between LPP siCtrl and LPP siStat3 treated animals were seen with regard to Cd8a and Iba1 (Figure 6b,c).

### 2.5. LPP siStat3 Reduces Stat3 Expression in the Core Region of the Tumor

The fact that we observed significantly improved survival and reduced tumor sizes following treatment with LPP/siStat3, but could not observe a significant Stat3-repression in whole tumor samples led us to hypothesize that Stat3 depletion might be locally restricted to the area surrounding the injection site. To further investigate this hypothesis, we performed laser-capture microdissection to excise the central area of the tumor (Figure 7a) that equals the site of LPP injection. In case a sample showed necrotic areas in this region, these were first excised and discarded. Next, we isolated RNA from excised tumor core regions and performed qRT-PCR. We used 6 samples per cohort for the analyses, and obtained quantifiable data from 4 samples for *Stat3* and the housekeeping gene *Tbp*, but only 3 for the additional housekeeping *Hprt* (Figure 7b). These analyses showed that mRNA expression of *Stat3* and the known Stat3 target gene *Mmp9* was significantly decreased following LPP siStat3 treatment. Additionally, *Snai1* showed a tendency for reduced expression. The expression of *Mmp2* remained unchanged. Due to the small sample sizes, this can be interpreted as a trend towards expressional suppression.

These results support our previously proposed hypothesis that LPP siStat3 can specifically target cancer cells to limit tumor growth in vivo. Despite the robust effects of LPP-formulated siStat3 on tumor growth, our results also indicate that only a small fraction of tumor cells can be reached using the current formulations and mode of application. These observations suggest that improvement of the bioavailability may allow further enhancement of therapeutic efficacies of this approach in vivo.

## 3. Discussion

Even after decades of research GBM still remains an incurable disease with one of the most dismal prognosis for cancer patients. The oncogenic transcription factor STAT3 represents a key signaling hub regulating many tumor-related processes including proliferation, migration, apoptosis-resistance, angiogenesis and immune evasion (reviewed in [12]). Furthermore, STAT3 expression is correlated with the highly aggressive mesenchymal subtype of glioma that is particularly resistant to conventional therapy [5,6] and is known to regulate stemness in glioma cells [6,23]. Previous findings from our and other groups had demonstrated that upstream pharmacological inhibition [22,42,43,44] and stable lentiviral depletion [17,21,23] of Stat3 provokes great antitumoral responses in vitro and improves survival of tumor-bearing mice in vivo.

To further advance the concept of targeting STAT3 in GBM, we decided to use a gene therapeutic approach aimed at interfering with STAT3 expression. To this end, we employed a novel approach by using siRNAs formulated in lipopolyplexes [32,37,45] that was investigated for its effects on cultured glioma cells and locally administered to syngeneic tumors of orthotopically transplanted mice that faithfully mimic all the hallmarks of human gliomas including high mitotic activity, focal necrosis surrounded by pseudopalisading cells and diffuse infiltration into the brain parenchyma [41]. Using this approach we could show that LPPs loaded with Stat3-siRNA inhibit proliferation of glioma cells in the absence of unspecific toxicity in vitro, significantly limit tumor growth in vivo and improve the overall survival of glioma-bearing mice. Our results also show that these antitumor effects were not associated with detectable differences in the numbers of infiltrating T-cells and microglia, suggesting that tumor cell-derived STAT3 represents the primary target of LPP-PEI-siRNA in vivo. The analyses of microglia were of particular interest because it was proposed that STAT3 is required for the activation of microglia [18,46] which usually acquire a pro-tumorigenic phenotype in glioma [18]. Since we did neither observe detectable differences in the number of microglia nor in their morphology, we concluded that Stat3 siRNA had no major influence on these brain-resident immune cells if supplied as LPP complexes, however.

Various approaches for siRNA delivery were also tested by other groups, e.g., by targeting Beclin1 using intranasal delivery of PEIs [47], by targeting Survivin upon stereotactic injection [28], by targeting Eg5 using siRNA delivered via an viral envelope [48], by using multifunctional surfactants as packaging reagents to target Hif-1α [49] or by dual targeting of EGFR and Akt2 using peptides [50]. This exemplifies that specific targeting of a molecular vulnerability of cancers can be exploited by various means. Accordingly, all these different approaches suffer, in principle, from the same drawbacks. These are, limited tissue distribution, securing the release in the target cells and avoidance of siRNA-degradation and/or capture by unintended cells (e.g., immune cells) or components of the extracellular matrix. Another recent approach using a compound drug consisting of an aptamer targeting PDGFbeta and an siRNA targeting Stat3 in glioma showed similar growth retardation in vitro and in vivo [19], although this study only used subcutaneous xenografts for the in vivo experiments. Interestingly, this study also indicated that a stable fraction of Stat3 protein remained present in the tumor cells that cannot be depleted using their approach, whereas reduction of Stat3 phosphorylation was very well possible. In line with these findings, we could observe depletion of phospho-Stat3 in vitro and in vivo, but did not observe depletion of total Stat3 following administration of LPP-siStat3 in vivo.

The major limitation of this study lies in the comparatively low perfusion range of siRNA achieved, providing an explanation why reduction of Stat3 expression (and its target genes) in the bulk tumor can be considered only as minor. Our data suggest that the LPP PEIs only reach a sub-fraction of the tumor cells, as seen by a clear tendency towards reduced *Stat3*-mRNA expression in the tumor core region. Due to the very low sample size (4 samples per group) a definitive answer cannot be inferred from this data. Based on the fact that (1) most samples after LPP siStat3 treatment have *Stat3* expression values below the mean of LPP siCtrl-treated animals and (2) the mRNA of the known Stat3 target gene *Mmp9* was significantly reduced after LPP siStat3 treatment, it can be deduced that LPP-mediated delivery of siRNA was successfully achieved. Additionally, based on the in vitro data we deduce that the siRNA can specifically and effectively deplete Stat3 expression, because we could successfully inhibit proliferation of Tu2449 cells and spheroids after LPP-mediated delivery of siStat3. Therefore we conclude that delivery to Tu2449 is possible even in more complex settings (i.e., spheroids). Hence, it is more likely that the limiting factor in our current approach is tissue dispersion of LPP complexes rather than siRNA specificity. Indeed, when exploring organotypic tissue slice cultures of intact tumor (xenograft) material with regard to nanoparticle tissue penetration using fluorophore-labeled siRNAs for microscopic evaluation of tissue penetrance, we did observe LPP to diffuse into the tissue only to a certain extent. LPP tissue penetration was found to be better than for polyplexes, reflecting lesser impairment of nanoparticle diffusion with reduced surface charge (zeta potential; [51]). However, in another in vivo system (non-tumorous mouse brain in an alpha-synuclein mouse model), even our polyplexes were found to distribute across the CNS down to the lumbar spinal cord after a single intracerebroventricular infusion, thus emphasizing a distribution sufficient for exerting biological effects [52].

The changes of Stat3-target genes like *Mmp9* and *Snai1* also provide mechanistic hints on how Stat3-depletion increases survival. In particular, the repression of *Mmp9* expression might reduce the migratory capacity of tumor cells due to their inability to modify the ECM. This could be further potentiated by repression of *Snai1*, which is a master regulator of EMT [15]. Furthermore, Snai1 has been shown to enhance an immunosuppressive phenotype in cancers [15]. LPP-siStat3-mediated *Snai1* depletion might therefore alleviate this phenotype and in turn make the tumors more accessible or amenable for immunotherapy. Future research should therefore be directed towards increasing intratumoral dispersion of siRNA in order to increase the knockdown efficiency. This could be achieved by further improving the existing complexation formulation for LPPs for example by reducing size and/or surface charges, or by enhancing the LPP concentration for being able to inject larger siRNA amounts in the maximum possible injection volume and/or development of alternative, increasingly sophisticated nanoparticle systems for delivery [26].

A different explanation for the limited Stat3-depletion could be changes in Stat3 turnover or stability. Accordingly, the half-life of Stat3 has been previously determined to be between 30 h in primary murine neurons to 90 h in hepatocytes [53] and around 50 h in Epstein-Barr-Virus-infected PBMCs [54] indicating pronounced variations between cell types. Considering that HeLa cells have a 50% turnover rate of ~60% of their entire proteome within 5 h [55] it might also be possible that the in vivo microenvironment stimulates the tumor cells to promote faster protein turnover, because of increased cell divisions. In this case, the treatment frequency every third day might be too low. However, the fact that we already observe strong increases in survival by 7 days (~25%) indicates that by continuous delivery of siRNA this issue can likely be resolved. One possibility to achieve continuous delivery could be to employ convection-enhanced delivery (CED) using implanted catheters. Accordingly, Chen et al. [56] could show that irinotecan-treatment using CED had the best survival responses in orthotopic, murine xenografts compared to conventional treatment approaches. In fact, the use of CED for treating glioma patients has also been tested in several clinical trials for the delivery of small molecules [57], antisense oligonucleotides [58] or siRNAs [59]. More recent reports also show the potential for multiplexed targeting using various siRNAs in established murine tumors [60]. Another study also reported the use of the so-called Cleveland Multiport Catheter (CMC) [61] which has already been tested in clinical trials (NCT02278510). The main advantage of the CMC is that it consists of four independent microcatheters and could therefore be employed to deliver multiple siRNAs and/or chemotherapeutics to more specifically target the vulnerabilities of a specific tumor. Despite these considerations, even the somewhat limited perfusion of Stat3-siRNA observed in our in vivo experiments with LPP therapy allowed to achieve impressive effects on tumor growth and overall survival, further supporting the suitability of STAT3 as an excellent target for GBM therapy. However, bearing in mind the enhanced colloidal stability of lipopolyplexes over their polyplex counterparts [35], the above mentioned approach of continuous delivery via direct infusion of the LPP into the tumor, using convection-enhanced delivery (CED) via an intratumoral catheter connected to a portable pump, also becomes feasible, especially because it is to be expected that siRNA stability or release kinetics would not be negatively affected by this approach.

## 4. Materials and Methods

### 4.1. Analysis of TCGA Data Sets

The publicly available TCGA datasets GBMLGG [62] and GBM [40] were accessed, analyzed and the plots were exported via the GlioVis portal (gliovis.bioinfo.cnio.es [63]). Comparison of Stat3 expression in different glioma grades was plotted based on the histology. The survival curves were derived from the GBM dataset of all GBM subtypes combined with a median cut-off, including confidence intervals.

### 4.2. Cells and Compounds

The murine GBM cell line Tu2449 [64] and double reporter cell line Tu2449-EGFP/Luc stably expressing EGFP and luciferase (Luc3) (kindly provided by Dr Alexander Wurm, Universitätsklinikum Leipzig, Leipzig, Germany), the human GBM cell line U-87 MG (U87, obtained from ATCC, Manassas, VA, USA) and Mz18 [65] were cultivated in DMEM Glutamax (high Glucose, Gibco/Thermo Fisher Scientific, Karlsruhe, Germany) with 10% heat-inactivated fetal calf serum and 1% penicillin/streptomycin (all from Gibco). Cultures were maintained in a humidified incubator at 37 °C and 5% CO_2_ and checked for mycoplasma contamination monthly by PCR Mycoplasma Test Kit II (AppliChem, Darmstadt, Germany) according to the manufacturer’s instructions.

### 4.3. siRNAs + Sequences

Chemically synthesized siRNAs were purchased from MWG (Ebersberg, Germany), Dharmacon/GE Healthcare (Lafayette, CO, USA) or Eurogentec (Seraing, Belgium), with sequences as follows: siLuc2/siCtrl (human cell lines): 5′-CGU ACG CGG AAU ACU UCG AdTdT-3′ (sense), 5′-UCG AAG UAU UCC GCG UAC GdTdT-3′ (antisense); siLuc3: 5′-CUU ACG CUG AGU ACU UCG AdTdT-3′ (sense), 5′-UCG AAG UAC UCA GCG UAA GdTdT-3′ (antisense); siEGFP: 5′-GCA GCA CGA CUU CUU CAA G-dTdT-3′ (sense), 5′-CUU GAA GAA GUC GUG CUG CdTdT-3′ (antisense), human STAT3 (siSTAT3-1: CGU UAU AUA GGA ACC GUA AdTdT (sense) and 5′-UUA CGG UUC CUA UAU AAC GdTdT (antisense), siSTAT3-2: GCC UCU CUG CAG AAU UCA AdTdT (sense) and UUG AAU UCU GCA GAG AGG CdTdG (antisense) or murine Stat3 (siStat3: 5′-GGC AUA UCG AGC CAG CAA AdTdT-3′ (sense) and 5′-UUU GCU GGC UGC AUA UGC CdTdT-3′ (antisense)), or negative control siRNAs (mouse: 5′-AAU CCG CUG UCG GCU GGA AdTdT-3′ (sense) and 5′-UUC CAG CCG ACA GCG GAU UdTdT-3′ (antisense)).

### 4.4. Polyplex and Lipopolyplex Preparation

Liposomes from 1,2-dipalmitoyl-sn-glycero-3-phosphocholine (DPPC; Avanti Polar Lipids, Alabaster, AL, USA) were prepared by the hydration/extrusion method as described previously [32]. For polyplex formation, siRNAs were complexed in HN buffer (150 mM NaCl, 10 mM HEPES, pH 7.4) with branched 4–12 kDa PEI F25-LMW [36] at PEI/siRNA mass ratio = 7.5:1 (N/P ratio 57) as previously described [27]. Lipopolyplexes (LPP) for in vitro experiments were prepared by incubating equal volumes of PEI/siRNA complexes with DPPC liposomes at a PEI/lipid mass ratio of 2.6 [32]. For this, 25 µL polyplex containing 0.8 µg siRNA, 6 µg PEI F25 and 25 µL DPPC liposomes comprising 16 µg lipid were properly mixed by pipetting and vortexing, and incubated for at least 1 h at room temperature. For in vivo application, lipopolyplexes were prepared by complexing 0.5 µg siRNA (200 µM), 3.75 µg PEI F25 (4.2 mg/mL) in 3 µL and mixed with 10 µg DPPC (5 mg/mL; 2 µL) to a total volume of 5 µL per injection.

### 4.5. qRT-PCR

For preparation of total cellular RNA from cell lines, phenol/chloroform extraction using 250 µL TRI-Reagent (Sigma-Aldrich, Munich, Germany) according to the manufacturer’s protocol was used. Employing the RevertAidTM H Minus First Strand cDNA Synthesis Kit (Fermentas, St. Leon-Roth, Germany), cDNA was transcribed from 800 ng RNA according to the manufacturer’s protocol. Quantitative PCR (qPCR) was performed in a StepOnePlus Real-Time PCR System (Applied Biosystems, Darmstadt, Germany) using the PerfeCTa SYBR^®^ Green FastMix ROX (Quantabio, Beverly, MA, USA). All procedures were conducted according to the manufacturers’ protocols with 4 µL cDNA (diluted 1:10 with nuclease free water), 1 µL primers (5 µM) and 5 µL SYBR Green master mix. After pre-incubation for 15 s at 95 °C, 40 amplification cycles followed: 10 s at 95 °C, 10 s at 55 °C and 10 s at 72 °C. A melting curve for PCR product analysis was recorded by rapid cooling down from 95 °C to 65 °C followed by incubation at 65 °C for 15 s prior to heating to 95 °C. Actin-specific primer sets were always run in parallel for each sample, to normalize for equal mRNA/cDNA amounts. Target levels were determined by the ΔΔCt method [66]. Following primers were used: human STAT3 fwd: GAG GAC TGA GCA TCG AGC A and rev: CAT GTG ATC TGA CAC CCT GAA, human Actin fwd: CCA ACC GCG AGA AGA TGA and rev: CCA GAG GCG TAC AGG GAT AG.

Pooled samples from laser-capture-microdissected formalin-fixed paraffin embedded (FFPE) tissues were subjected to RNA-Isolation using the Arcturus^®^ Paradise^®^ PLUS FFPE RNA Isolation Kit (Thermo Fisher) and cDNA-synthesis using SuperScript IV Vilo (life technologies/Thermo Fisher) per the manufacturer’s instructions. Briefly, the RNA was eluted in 12 µL DEPC-treated H_2_O and used entirely for the cDNA-synthesis without prior RNA quantification. The qPCR was performed using 20× Taqman Probes (Applied Biosystems, Darmstadt, Germany) and 2× Fast-Start Universal Probe Master Mix (Roche) and 2 µL of cDNA on a StepOne Plus System (Applied Biosystems) using the standard setting. The gene expression values were normalized to TATA-Box binding protein (Tbp) or Hypoxanthine-guanine phosphoribosyltransferase (Hprt) by calculating the mean Ct value of both housekeeping gene and using this value as reference for further normalization using the ΔΔCt method [66]. The following Probes were used: Hprt (Mm00446966_m1); Mmp2 (Mm00439506_m1); Mmp9 (Mm00442991_m1); Snai1 (Snail, Mm01249564_g1); Stat3 (Mm01219775_m1); Tbp (Mm01219775_m1).

### 4.6. Western Blotting

Preparation of protein lysates from Tu2449 cells [67] and mouse brain tumors [68] were prepared as described. SDS-PAGE and Western Blotting was performed as described [67]. Membranes were blocked in 5% BSA/TBS-Tween20 (TBS-T) for 1 h at room temperature. Antibodies were incubated at 4 °C overnight in 5% BSA/TBS-T, secondary goat anti-mouse or goat anti-rabbit (dilution 1:10,000, Li-Cor Biosciences, Bad Homburg, Germany) were incubated at room temperature for 1 h and detection was achieved using a LI-COR Odyssey reader (LI-COR Biosciences, Bad Homburg, Germany).

For protein analysis of human cell cultures, cells were seeded and transfected in 24-well plates as described above. After 72 h, medium was removed and cells were washed once with PBS. 80 µL RIPA lysis buffer (50 mM Tris (pH 7.4), 150 mM NaCl, 1% Triton X-100, 0.5% sodium deoxycholate, 0.1% SDS, 2.5 mM sodium pyrophosphate, 1 mM EDTA, 10 mM NaF, Protease Inhibitor Cocktail Set III (EDTA-free, Merck, Darmstadt, Germany) was added per well and plates were incubated on ice for 10 min. The suspension was transferred to Eppendorf tubes. After centrifugation (10,000 rpm, 4 °C, 10 min), the supernatant was transferred to a new Eppendorf tube.

Using the Bio-Rad DCTM Protein-Assay (Bio-Rad, Munich, Germany), protein concentration was determined according to manufacturer’s protocol. 4× loading buffer was added (0.25 mM Tris-HCl, pH 6.8, 20% glycerol, 10% beta-mercaptoethanol, 8% SDS, 0.08% bromophenol blue) to yield a 1× concentration and 20 µg protein was loaded onto 10% polyacrylamide gels. Proteins were separated by SDS-PAGE and transferred to a 0.45 µm ImmobilonTM-P Transfer PVDF Membrane (Millipore, Burlington, MA, USA). Membranes were blocked with blocking buffer (5% (*w*/*v*) milk powder in TBST (10 mM Tris-HCl, pH 7.6, 150 mM NaCl, 0.1% Tween 20) for 30 min. After washing in TBST, membranes were incubated with primary antibodies diluted in 3% milk powder (*w*/*v*) in TBST: anti-human STAT3 (Thermo Fisher Scientific), anti-Actin (Cell signaling Technologies (CST (Frankfurt am Main, Germany)), or anti-Vinculin (Sigma-Aldrich) overnight at 4 °C. Then, blots were washed in TBST and incubated for 1 h with horseradish peroxidase-coupled goat anti-rabbit IgG (CST) or horseradish peroxidase-coupled goat anti-mouse IgG (Thermo Fisher Scientific) in 3% milk powder (*w*/*v*) in TBST before washing again. Bound antibodies were visualized by enhanced chemiluminescence (ECL kits: SignalFireTM (CST) or SuperSignal^®^ West Femto (Thermo Fisher Scientific)).

### 4.7. Antibodies

The following antibodies and dilutions were used: Actin (CST, 13E5, 1:1000 in 3% milk); Cluster of Differentiation 4 (CD4); Affymetrix eBioscience/Thermo Fisher; clone 4SM95; 14-97664; 1:100 (IHC); CD8a (Synaptic Systems, Göttingen, Germany) 361003; 1:500 (IHC); Gapdh (Calbiochem, #CB1001, Darmstadt, Germany) 1:20,000; Iba1 (Wako, 019-19741); 1:1000 (IHC); Ki67 (SP6, Thermo Fisher, MA5-14520); 1:100 (IHC); Stat3 (Santa Cruz Biotechnology (Heidelberg, Germany)), c-20; sc-482; 1:200 (WB); Stat3 (phospho Tyr705; CST Technologies (CST (Frankfurt am Main, Germany)); D3A7; 9145S; 1:1000 for Western Blot (WB) and 1:500 for Immunohistochemistry (IHC)); STAT3 (Thermo Fisher, 9D8, 1:5000 in 3% milk); Vinculin (Sigma Aldrich, hVIN-1, 1:2000 in 3% milk).

### 4.8. Nanoparticle Characterization

Particle sizes and zeta potentials were determined by photon correlation spectroscopy (PCS) and phase analysis light scattering (PALS), using a Brookhaven ZetaPALS system (Brookhaven Instruments, Holtsville, NY, USA). The data were analyzed using the manufacturer’s software and applying a viscosity and refractive index of pure water at 25 °C. For size determination, the complexes were analyzed in five runs with a run duration of 1 min. Results are expressed as intensity weighted mean diameter from different experiments. Zeta potentials were measured in ten runs, with each run containing ten cycles, and applying the Smoluchowski model. Additionally, hydrodynamic diameters of the nanoparticles were determined by nanoparticle tracking analysis (NTA), using a NanoSight LM 10 HS apparatus (Malvern) equipped with a 640 nm sCMOS camera and a temperature controlled sample chamber, as described previously [45,69].

### 4.9. Cell Transfection

EGFP and luciferase knockdown experiments were performed in 24-well plates. The day before, Tu2449-EGFP/Luc cells were seeded at a density of 30,000 cells per well in 0.5 mL fully supplemented medium. Polyplexes and lipopolyplexes comprising 0.8 µg siRNA per well were prepared as described above, containing siRNAs against the targets EGFP or luciferase, or the negative control siRNA (see above Section 4.3).

For transfection using INTERFERin^TM^ (Polyplus, Illkirch, France), 2 × 10^2^ cells were seeded in a 96-well plate (proliferation assay; U87 or Mz18 cells) or 2 × 10^5^ cells in a 24-well plate (cytotoxicity assay; Tu2449 cells), respectively. Cells were incubated under standard conditions unless stated otherwise. 10 nM siRNA were transfected according to the manufacturer’s protocol and incubated for the indicated periods of time.

### 4.10. Determination of Knockdown Efficacies

Luciferase activities were determined in the double reporter cell line Tu2449-EGFP/Luc 96 h post transfection using the Beetle-Juice Kit (PJK, Kleinblittersdorf, Germany). The medium was aspirated and 300 µL lysis buffer (Promega, Mannheim, Germany) was added, prior to incubation for 30 min at RT. In a test tube, 25 µL luciferin substrate was mixed with 10 µL cell lysate and luminescence was immediately measured in a luminometer (Berthold, Bad Wildbad, Germany).

The EGFP knockdown was quantitated after 96 h by flow cytometry. Cells were trypsinized and centrifuged for 3 min at 3000 rpm followed by a washing step with 1% BSA in PBS (1 mL), and finally resuspended in 0.5 mL of the same buffer. The cells were measured in an Attune^®^ Acoustic Focusing Cytometer (Life Technologies, Darmstadt, Germany) with appropriate instrument settings. 20,000 events were gated and analyzed using the Attune^®^ software (V2.1.0). The data are presented as mean fluorescent intensities.

### 4.11. Cell Proliferation and Viability Assays

For proliferation assays, U87 or Mz18 cells were seeded and transfected in 96-well plates as described above. The number of viable cells was determined at the time points indicated using a colorimetric assay. Briefly, 50 µL of a 1:10 dilution of Cell proliferation Reagent WST-1 (Roche Molecular Biochemicals, Mannheim, Germany) in serum-free medium was added to the cells after aspirating the media. Cells were incubated for 1 h at 37 °C and absorbance at 450 nm was measured in an ELISA reader.

Acute cell damage upon transfection was analyzed in Tu2449 cells by the lactate dehydrogenase (LDH) release assay, using the Cytotoxicity Detection Kit (Roche, Mannheim, Germany) according to the manufacturer’s protocol. Briefly, conditioned medium from knockdown experiments was collected after 24 h. Conditioned medium from untreated cells served as negative control, and for the determination of maximum LDH release (100% value), cells were lysed by adding Triton X-100 to final concentration of 2% into the medium. In a 96 well plate, 50 µL sample medium was mixed with 50 µL reagent mix and incubated for 30 min in the dark. The reaction was stopped with 50 µL 1 M acetic acid and the absorption at 490 nm and 620 nm as reference filter was measured in an ELISA reader. Fresh fully supplemented medium and reagent mix served as background and was subtracted from all values. Cytotoxicity values are shown in percent of the maximum value.

### 4.12. Cell Cycle Analysis

Cell cycle distribution was assessed using flow cytometry. Cells were seeded in 24-well plates and transfected as described above. 48 h after transfection, cells were trypsinized, transferred to Eppendorf tubes and centrifuged (1500 rpm, room temperature, 5 min). The supernatant was aspirated and the cell pellet was resuspended in ice cold 70% ethanol and incubated for at least 1 h at 4 °C for fixation. After centrifugation (1500 rpm, room temperature, 5 min) and discarding the supernatant, cells were resuspended in an RNase A solution (50 µg/mL RNase A in PBS) and incubated for 30 min at 37 °C. Propidium iodide solution was added to yield a 50 µg/mL concentration. Cell cycle distribution was analyzed in the BL2 channel using an Attune^TM^ Flow Cytometer.

### 4.13. Spheroid Assay

To generate spheroids, 3 × 10^3^ Tu2449 cells were seeded in a 96-well U-shaped ultra-low attachment plate and incubated under standard conditions. Using SAINT-sRNA transfection reagent (Synvolux, Leiden, The Netherlands), 90 nM siRNA were transfected according to the manufacturer’s protocol.After 10 days, spheroid size was documented and measured using a Celigo Imaging Cytometer (Nexcelom Bioscience, Lawrence, MA, USA) and the corresponding software.

### 4.14. Animal Experiments

Intracranial implantation of tumor cells into the right striatum of B6C3F1 mice (Envigo, Huntingdon, UK; average weight 25 g) was performed as described [21]. Briefly, 10,000 viable Tu2449 in 1 µL were injected with a flow rate of 0.5 µL/min at a depth of 3 mm 1.5 mm posterior and 2 mm lateral to the bregma. 42 mice were implanted with tumor cells and one week after implantation the animals were randomly divided into two groups; one received LPP siCtrl treatment, the other LPP siStat3 treatment. Application of LPPs (equivalent to 0.5 µg siRNA in 5 µL which, due to limitations regarding the total lipopolyplex volume, was the maximum possible amount in this experimental setting) was performed using the same drill hole in the skull as for tumor cell implantation every third day one week after tumor cell implantation. 21 days after tumor implantation 10 mice per group were sacrificed; of those, 8 mice were used for histological analysis and two for molecular analyses. Those tumors were divided and one half was immediately frozen on dry ice and used for protein isolation, the other half was stored in RNAlater (Qiagen, Hilden, Germany) and used for RNA isolation. All intracranial injections were performed on anesthetized animals and treatment was done unblinded. Histoligical analysis showed that in the LPP siCtrl and LPP siStat3 group three and five tumors could not be established. Of those, one and two tumors were planned for histological assessment after 21 days. These mice were removed from the survival-analysis. Animals at the collection time point or those that showed neurological symptoms related to tumor burden were euthanized by cervical dislocation after they received a lethal injection of anesthetic. Hereafter the brains were immediately removed, rinsed in PBS and fixed for 2–7 days in 4% PFA (Chemcruz/Santa Cruz). Prior to paraffin embedding the fixed brains were cut coronally in 2–3 mm thick section and the resulting sections were aligned horizontally. Embedding in paraffin was performed according to standard paraffin embedding procedures known from routine pathology. Sildes were first stained with hematoxylin and eosin. Immunohistochemistry was performed on an automated staining system Bond III (Leica Microsystems, Wetzlar, Germany) using standard protocols. All animal experiments were approved by the local administrative council (Regierungspräsidium Darmstadt, FK/1011).

### 4.15. Determination of Tumor Size

The area of tumors was determined by measuring the longest distance (a) between tumor borders and the respective orthogonal line (b) from all tumor sections visible from H&E stained tumor sections. Using the formulas for ellipses the area (A) was calculated:A = π × 0.5 × a × b

If a brain contained multiple tumor sections those values were summed to account for the depth of the tumor. If multiple sections from one mouse were available the area, indicative for the tumor center, was chosen.

### 4.16. Laser-Capture Microdissection

Laser capture microdissection (LCM) was performed of 10 µm thick, serial sections mounted on UV-irradiated MMI membrane slides (MMI, Eching, Germany) and collected on MMI Isolation caps with diffuser caps (MMI). Prior to microdissection, the sections were dried at 37 °C for 1 h and rehydrated (2 times 10′ xylene, 99%, 96%, 70% EtOH, 30′′ each) and the first and last section of the series were stained with hematoxylin and eosin, dehydrated and mounted using Pertex (Leica, Wetzlar, Germany). LCM was performed using a laser capture microscope (Oylmpus IX71 equipped with MMI CellCamera und MMI CellCut) operated by the software MMI CellTools (v. 4.0).

### 4.17. Statistics

All statistical analysis were performed using GraphPad Prism 7 (GraphPad Software, La Jolla, CA, USA). Kaplan-Meier-Survival curves were analyzed using log-rank test, all other experiments using two-tailed T-Test, with assuming equal SD and without corrections for multiple comparisons. Cell cycle measurement was analyzed by two-way-ANOVA of matched samples with Dunnett’s multiple comparison test against siCtrl treated cells. Unless stated otherwise, all samples are presented as Box-Plots (min-to-max), where the horizontal line depicts the median value and the “+” symbol depticts the calculated mean.

## 5. Conclusions

In conclusion, we provide first proof-of-concept evidence that intratumoral targeting of oncogenes like Stat3 using lipopolyplexsosomal siRNA for the treatment of brain tumors is feasible and leads to improved overall survival in a murine, syngeneic, orthotopic transplantation model. Future development of nanoparticle systems and the modes of their application will hopefully enhance the restricted tissue distribution achieved in this study, thereby further improving the tumor targeting capacity of siRNA. These efforts may lead to the development of new and highly specific treatments selectively targeting the expression of oncogenes such as STAT3, an avenue recently opened in a clinical trial focusing on another gene target in brain tumors (NCT03020017).

## Figures and Tables

**Figure 1 cancers-11-00333-f001:**
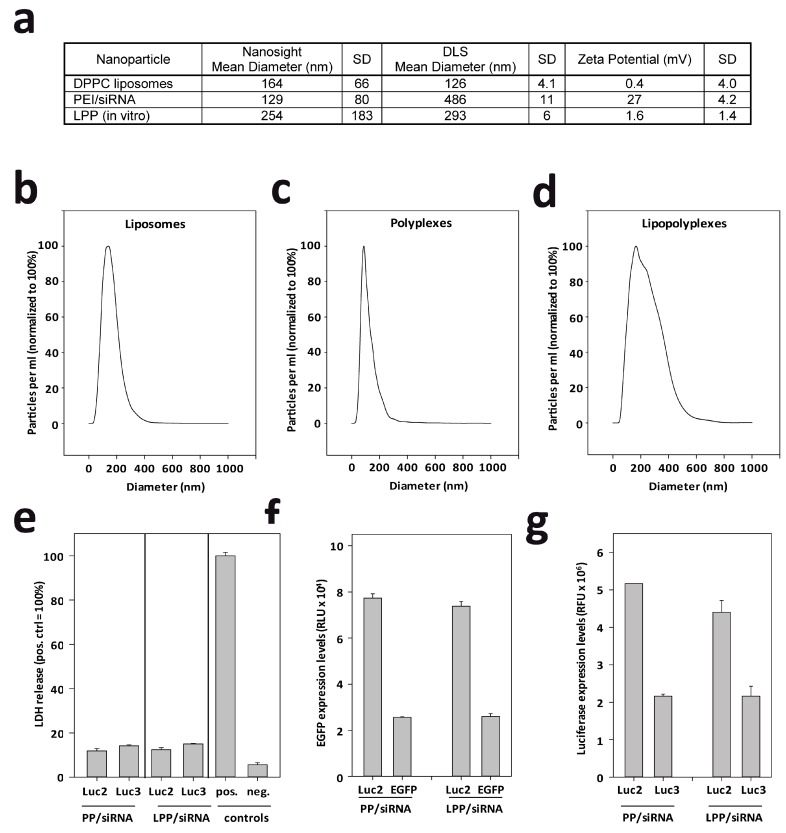
(**a**) Compilation of nanoparticle sizes and zeta potentials, as determined by Zetasizer and Nanosight. (**b**–**d**) Original diagrams of size measurements by Nanosight. (**e**–**g**) Biological properties of nanoparticles. (**e**) Determination of cytotoxicity by LDH release assay. (**f**,**g**) Determination of knockdown efficacies in EGFP/Luciferase (Luc3) Tu2449 reporter cells. Knockdown of (**f**) EGFP and (**g**) luciferase (Luc3) is shown. For (**a**–**d**) two independent samples were measured 10 times; (**e**) was performed twice in 4 biological replicates and (**f**,**g**) were performed twice in biological duplicate; shown are the summaries of all experiments.

**Figure 2 cancers-11-00333-f002:**
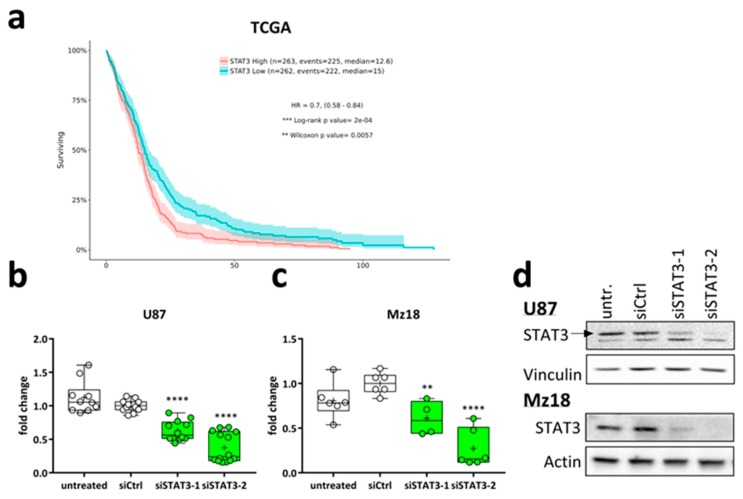
(**a**) Kaplan-Meier-Survival Plot from TCGA dataset GBM [40] showing that high STAT3 expression is associated with shorter survival; (**b**,**c**) qRT-PCR from (**b**) U87 and (**c**) Mz18 human glioma cell lines after transfection with control siRNA (siCtrl) or two siRNAs against STAT3 (siSTAT3-1 and siSTAT3-2). STAT3-expression was normalized to Actin as housekeeper and siCtrl-transfected cells as control sample using the ΔΔCt-method. The data are presented as box-plots (min-to-max) with all samples displayed as circles; the horizontal line in the box depicts the median value, the plus-symbol the mean. (**d**) Western Blot of U87 and Mz18 after transfection as in (**b**,**c**) after transfection of siCtrl, siSTAT3-1 or siSTAT3-2. (**e**–**h**) Proliferation (WST-1) assays of the human glioma cell lines (**e**) U87 and (**f**) Mz18, using INTERFERin and the two different siSTAT3 for comparison, and in the murine glioma cell line Tu2449 after transfection with (**g**) INTERFERin^TM^ or (**h**) LPP. The data in (**e**–**g**) are presented as mean +/− SEM; the data in (**h**) are presented as Box-Plots (min-to-max) with all samples displayed. (**i**) Western Blot of Tu2449 cells 96 h after transfection with 150 pmol LPP siCtrl or LPP siStat3. (**b**,**c**) shows the summary of at least three independent experiments performed in biological duplicates; (**d**) was performed twice; (**e**,**f**,**h**) were performed three (**g**) two times in biological triplicates; (**i**) was performed three times. **: *p* < 0.01; ***: *p* < 0.001 and ****: *p* < 0.0001 compared to siCtrl treatment.

**Figure 3 cancers-11-00333-f003:**
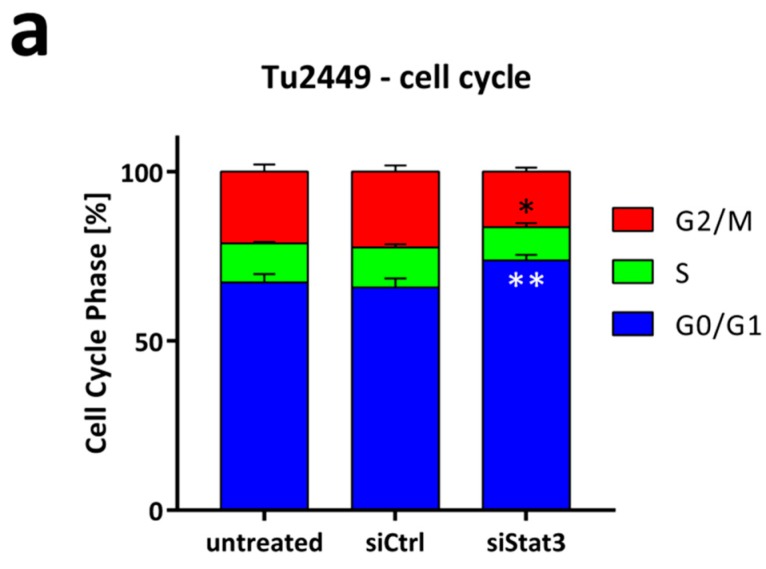
(**a**) Cell cycle analysis of Tu2449 cells after transfection of siCtrl or siStat3. A significant increase in the percentage of cells in the G1 phase suggests antiproliferative properties of Stat3-depletion. (**b**,**c**) Sphere growth assay of Tu2449 after transfection of siCtrl or siStat3. Three representative spheres for each condition are presented in (**b**) the quantification after 10 days of sphere growth is shown. Scale bar: 500 µm. (**c**) as box-plots (min-to-max) with all samples displayed as circles; the horizontal line in the box depicts the median value, the plus-symbol the mean. *: *p* < 0.0.05; **: *p* < 0.01 and ****: *p* < 0.0001 compared to siCtrl treatment. A representative experiment of at least three independent repetitions is shown.

**Figure 4 cancers-11-00333-f004:**
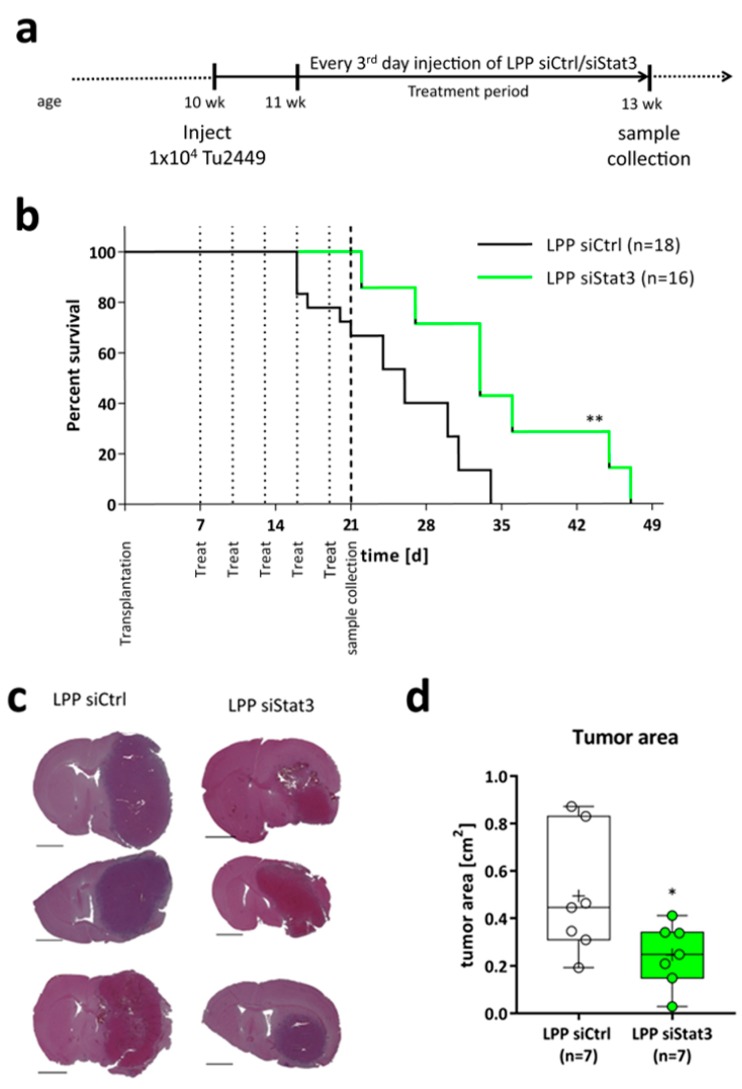
(**a**) Schematic presentation of the in vivo experiments. Briefly, 42 mice were injected with 10,000 Tu2449 cells into the right striatum. After 7 days the mice were randomly divided into two groups of 21 animals each and intracranial treatment was performed every third day. After 21 days 10 mice were euthanized for histological and molecular analysis. The remaining mice were monitored for long-term survival. Mice without established tumors were excluded from the analysis (3 (1 for histology; 2 for survival) for LPP siCtrl; 5 (1 for histology; 4 for survival) for LPP siStat3) (**b**) Kaplan-Meier-Survival Plot of B6C3F1 mice after implantation of Tu2449 and treatment with LPP siCtrl or LPP siStat3. The vertical dotted lines depict the treatment days, the vertical dashed line the day of sample collection. LPP siStat3 treatment significantly increases median survival from 26 days to 33 after LPP siCtrl and LPP siStat3 treatment respectively (**c**) Three representative brain sections with tumors 21 days after tumor cell implantation and treatment with LPP siCtrl (left side) or LPP siStat3 (right side); scale bar: 2 mm. (**d**) Box-Plots (min-to-max) with all samples displayed as circles; the horizontal line in the box depicts the median value, the plus-symbol the mean of tumor areas 21 days after tumor cell implantation after treatment with LPP siCtrl or LPP siStat3. *: *p* < 0.05; **: *p* < 0.01 compared to LPP siCtrl.

**Figure 5 cancers-11-00333-f005:**
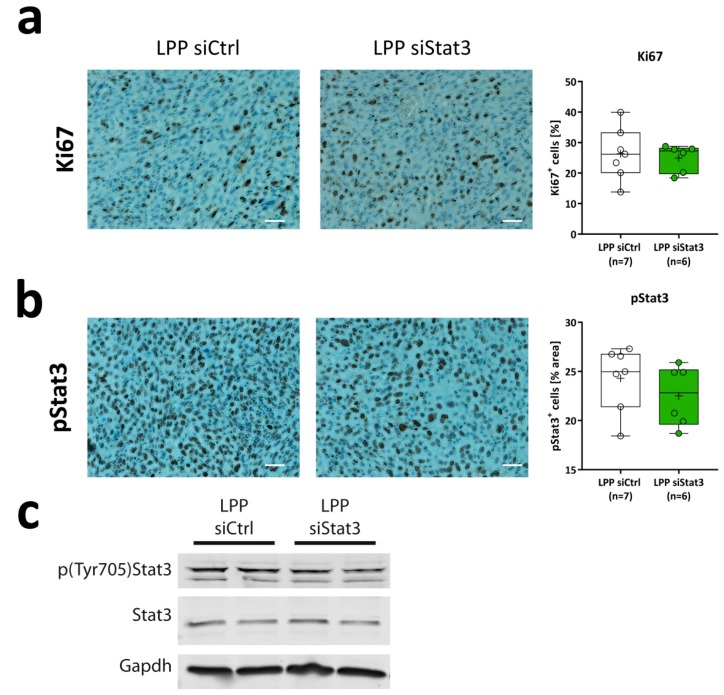
(**a**,**b**) Immunohistochemical analysis of (**a**) Ki67 and (**b**) tyrosine-phosphorylated Stat3 (pStat3) expressing tumor cells 21 days after implantation and treatment with LPP siCtrl and LPP siStat3. One representative picture is shown for each staining and treatment and the quantification is presented as box-plots (min-to-max) with all samples displayed as circles; the horizontal line in the box depicts the median value, the plus-symbol the mean of tumor areas 21 days after tumor cell implantation after treatment with LPP siCtrl or LPP siStat3. Scale bar: 200 µm. (**c**) Western Blot of whole-tumor lysates from two mice each after treatment with LPP siCtrl or LPP siStat3; Gapdh was used as the loading control.

**Figure 6 cancers-11-00333-f006:**
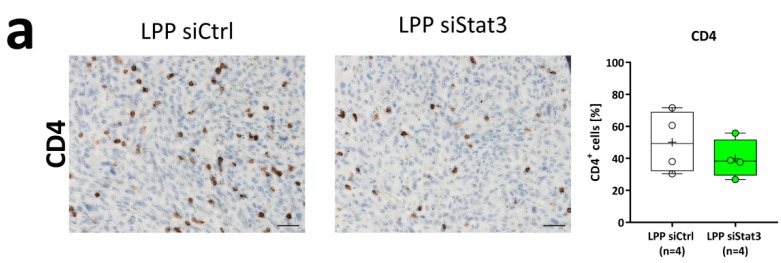
(**a**–**c**) Immunohistochemical analysis of (**a**) CD4, (**b**) CD8a and (**c**) Iba1 expressing tumor cells 21 days after implantation and treatment with LPP siCtrl and LPP siStat3. One representative picture is shown for each staining and treatment and the quantification is presented as box-plots (min-to-max) with all samples displayed as circles; the horizontal line in the box depicts the median value, the plus-symbol the mean value. scale bar: 200 µm.

**Figure 7 cancers-11-00333-f007:**
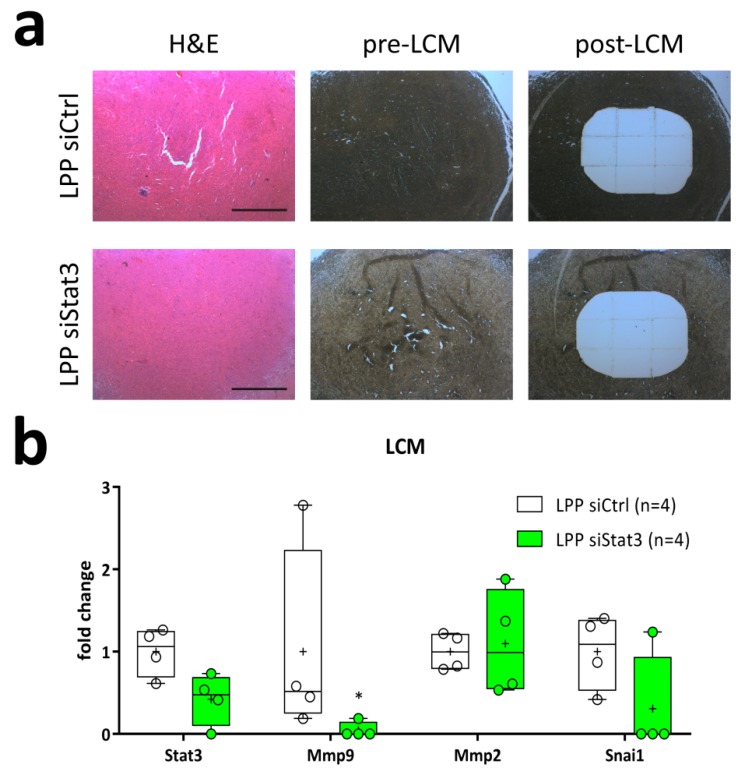
(**a**) Representative histology of tumor centers stained with H&E (left panel) and of unlabeled sections before (middle panel, pre-LCM) and after (right panel, post-LCM) microdissection of the central part of the tumor. Scale bar: 400 µm. (**b**) Taqman-based gene expression analysis normalized using the housekeepers *Hprt* and *Tbp* depicted as box-plots (min-to-max) with all samples displayed as circles; the horizontal line in the box depicts the median value, the plus-symbol the mean value. *: *p* < 0.05 compared to LPP siCtrl.

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
