# Peer review of "Therapeutic Targeting of Stat3 Using Lipopolyplex Nanoparticle-Formulated siRNA in a Syngeneic Orthotopic Mouse Glioma Model"

_cancers, 2019, doi:10.3390/cancers11030333_

Reviewer 1 Report

This manuscript describes the in vitro and in vivo application of siRNA against STAT3 using a polycation/lipid based delivery vectors. The in vitro part appears clear and concise, showing a robust target knockdown accompanied by reduced tumor cell proliferation. Nevertheless, the in vivo experiment remains incomplete. Although the model (orthotopic, i.e. intracranial tumor implantation, syngeneic model>) appears of high quality, the results obtained are not convincing. For me, the differences in tumor size (rather area of the (most middle?) histological section) are on the edge of significance (2/7 tumor in the control group are clearly bigger, 1/7 tumors in the siSTAT3 group is considerably smaller), same applies to the increase in survival time. The in vitro results suggest that STAT3 knockdown can indeed act antiproliferative. In their previous studies the authors convincingly showed that STAT3 knockdown in vivo is indeed acting antitumoral, also in glioblastoma. In my view, the in vivo knockdown in this study is simply not efficient enough, and I guess that it is the limited in vivo knockdown efficiency achieved with this formulation. In terms of distribution/diffusion, it would not see so much limitation, as there have been in total five separate injections per tumor, which should in principle ensure that all tumor areas are covered. It would be a pity not to publish such a study involving this high number of experimental mice, but for me the current status of the in vivo part remains incomplete. A solution to this problem could be to pinpoint the local knockdown in the tissue either by doing immunohistochemical analyses for STAT3 (as shown in Fig 4B, but for the whole section), or, alternatively, analyzing the distribution pattern in the siRNA delivered (e.g. using a labelled probe, FL or radioactive). This could of course be done with the existing samples. As the authors also state in the discussion, the knockdown efficiency in vivo with the currently used delivery vector is limited. Hence, being able to pinpoint the limiting steps (tissue distribution OR efficacy of knockdown) would really clarify a lot.

Minor points:

Fig 1: Figurelegend appears truncated/incomplete, same for x-axis labels (e.g. what is Luc 2, Luc 3? 1f and 1g: x-axis label is missing (scrambled vs a-EGFP siRNA??)

Author Response

This manuscript describes the in vitro and in vivo application of siRNA against STAT3 using a polycation/lipid based delivery vectors. The in vitro part appears clear and concise, showing a robust target knockdown accompanied by reduced tumor cell proliferation. Nevertheless, the in vivo experiment remains incomplete. Although the model (orthotopic, i.e. intracranial tumor implantation, syngeneic model>) appears of high quality, the results obtained are not convincing. For me, the differences in tumor size (rather area of the (most middle?) histological section) are on the edge of significance (2/7 tumor in the control group are clearly bigger, 1/7 tumors in the siSTAT3 group is considerably smaller), same applies to the increase in survival time. The in vitro results suggest that STAT3 knockdown can indeed act antiproliferative. In their previous studies the authors convincingly showed that STAT3 knockdown in vivo is indeed acting antitumoral, also in glioblastoma. In my view, the in vivo knockdown in this study is simply not efficient enough, and I guess that it is the limited in vivo knockdown efficiency achieved with this formulation. In terms of distribution/diffusion, it would not see so much limitation, as there have been in total five separate injections per tumor, which should in principle ensure that all tumor areas are covered. It would be a pity not to publish such a study involving this high number of experimental mice, but for me the current status of the in vivo part remains incomplete. A solution to this problem could be to pinpoint the local knockdown in the tissue either by doing immunohistochemical analyses for STAT3 (as shown in Fig 4B, but for the whole section), or, alternatively, analyzing the distribution pattern in the siRNA delivered (e.g. using a labelled probe, FL or radioactive). This could of course be done with the existing samples. As the authors also state in the discussion, the knockdown efficiency in vivo with the currently used delivery vector is limited. Hence, being able to pinpoint the limiting steps (tissue distribution OR efficacy of knockdown) would really clarify a lot.

Answer: We thank the reviewer for his appraisal of the in vitro part and the applied syngeneic mouse model. Regarding the efficacy of the siRNA in vivo, we would like to point out that despite the technical limitations of our approach (as previously discussed in the manuscript and below), tumor sizes and overall survival were clearly significant between the groups, with the normal variations in the respective data sets expected for such an experiment. In fact, we achieved an increase in survival time of ~25% (from 26 to 33 days respectively) which would be a definitive improvement in regard to GBM. Of course, future studies should employ higher numbers of animals for the histological examination given the variability of responses. However, considering that also LPP siCtrl-treated animals show similar variations in terms of overall tumor size, we consider our data to be within an acceptable range. Still, we agree that the in vivo knockdown efficiency as shown in Fig. 4 could be further improved. Based on our in vitro data we hypothesize that insufficient intratumoral distribution rather than the lack of siRNA efficacy is the limiting factor here, because the knockdown achieved in vitro using the same siRNA-sequences is quite convincing. On an informative note, we discussed all histological sections in detail with our well-experienced neuropathologists and also discussed how to perform the quantifications most objectively. Accordingly, we selected five random vision fields per slide not being located at the tumor border or containing necrotic regions that sometimes occur in very large tumors. Hence, we chose not to present whole-tumor sections, because without careful examination, some areas within the tumor might be misinterpreted and we thoroughly made sure to exclude these areas for our analysis.

Although technically difficult for several reasons (repeated injections at different timepoints, tumor heterogeneity, siRNA stability), we tried to pinpoint the local knockdown of STAT3 using the laser-capture-microdissection approach. This approach is of course much more complicated and time-consuming than IHC-stainings, but we are confident that it offers much more reliability and insight, because we selectively analyze viable tissue at the center of the tumor. Compared to bulk tumor lysates (Fig. 4C), dissection of the tumor center shows a trend for reduced levels of Stat3, and for two known targets of Stat3 (Mmp9, and Snai1). Admittedly, due to the low sample size this conclusion requires some caution. Nonetheless, as a proof-of-concept-study, the approach used here may be of great scientific and clinical importance.

We agree with the reviewer that an analysis of the tissue distribution would possibly add further proof, but due to time restrictions, we unfortunately could not perform the RNA hybridisation experiment suggested. As we already used up most of the FFPE tissue, it is doubtful whether we would be able to perform these experiments at all, especially considering that staining against RNA requires fresh samples that were not exposed to oxygen for extended periods of time, because of RNA degradation. Even the application of a labeled siRNA in vivo would not provide definitive proof of tissue dispersion and/or cellular uptake, because it is also possible that these labels can be detached from the siRNA and lead to misinterpretations of siRNA distribution. Based on the criticism of the reviewer, we have added further discussion of these findings in the manuscript and moderated our conclusions, especially considering the data from the LCM-approach (lines 300 to 305).

Minor points:

Fig 1: Figurelegend appears truncated/incomplete, same for x-axis labels (e.g. what is Luc 2, Luc 3? 1f and 1g: x-axis label is missing (scrambled vs a-EGFP siRNA??)

Answer: The Reviewer is correct that some x-axis labels were truncated in the manuscript. We assume that this must have happened during file conversion since this problem did not appear in the original figures. In the revised manuscript we now provide the full-length x-axis labels.

Also, the Reviewer correctly pointed out that the difference between siRNAs Luc2 and Luc3 had not been properly explained. These siRNAs target the Promega Luciferase vectors pGL2 and pGL3, respectively. In the reporter cell lines used in this study, pGL3 is stably integrated; thus, siRNA Luc3 is the specific siRNA while Luc2 served as negative control (Fig. 1g). In Fig. 1f siRNA EGFP is the specific siRNA, with again siLuc2 serving as negative control. In Fig. 1e, both siRNAs (Luc2 and Luc3) are shown simply to make the point that (absence of) toxicity is independent of the siRNA sequences, i.e., identical results are obtained for both siRNAs. By making the appropriate additions, this information is now given in the revised manuscript.

Also, we revised the Figure legend and added missing information for clarification.

Reviewer 2 Report

There are three major comments we concern in your study.

1.     There is less description of siRNA vehicle when employing the liposome in vehicle design such as the Rational of liposome added in the siRNA vehicle, the Stability of the siRNA vehicle with or without liposome, Release Profile of the siRNA with or without liposome, cytotoxicity of the siRNA with or without liposome?

2.     The strategy of drug delivery by every third day one week would be a big challenge in further treatment. Even the strategy of treatment would apply in CED, the property of stability and release kinetics of siRNA would be a challenge in the treatment.

3.     There is less discussion relating immune study and signal pathway. What is the significance of those gene regulation and immune suppression? 

Author Response

There are three major comments we concern in your study.

1.     There is less description of siRNA vehicle when employing the liposome in vehicle design such as the Rational of liposome added in the siRNA vehicle, the Stability of the siRNA vehicle with or without liposome, Release Profile of the siRNA with or without liposome, cytotoxicity of the siRNA with or without liposome?

Answer:

We agree with the reviewer that the description of this part was rather short and now added relevant background information as follows:

More recently, this [PEI-based siRNA complexation] was extended towards combinations with liposomes, leading to lipopolyplexes (LPP) that combine properties of both components (Ewe, Panchal et al. 2017). For gene delivery, LPP comprising phospholipids had been shown previously to display strongly reduced surface charges, enhanced transfection efficiencies, decreased cytotoxicity and high colloidal stability as compared to their unmodified complex counterparts (Heyes, Palmer et al. 2007, Ko, Kale et al. 2009). Likewise, we were able to demonstrate for siRNA delivery that DPPC-based LPP without co-lipids displayed very good transfection efficiency and further enhanced biocompatibility in cell culture. Intracellular siRNA release was not impaired by the liposomal component. Due to strongly reduced surface charges, storage stabilities under various conditions were markedly increased by protecting the LPP from aggregation (Ewe, Schaper et al. 2014). Importantly, these favorable properties also translated into therapeutic in vivo efficacy upon systemic injection into tumor xenograft-bearing mice (Ewe, Panchal et al. 2017), thus providing the basis for now switching to another model and another mode of administration.

2.     The strategy of drug delivery by every third day one week would be a big challenge in further treatment. Even the strategy of treatment would apply in CED, the property of stability and release kinetics of siRNA would be a challenge in the treatment.

Answer: We agree that a treatment every third day would be challenging from a clinical perspective. The rationale for this scheme was based on the in vitro findings in Tu2449 cells that after 72h we observe a slight and after 96h a pronounced knockdown of Stat3, respectively (compare Fig. 2i). Hence, we reasoned that a treatment every third day will most likely result in continuous depletion of Stat3 expression and high (enough) effective concentrations of the siRNA during the treatment period. Nonetheless it might very well be possible (from a technical point of view) to perform a clinically achievable treatment scheme of three times in 5 days followed by two days of pause (i.e. treatment on Monday, Wednesday and Friday).

It should be pointed out, however, that the sustained local delivery of nucleic acids via direct infusion of the compound (e.g., the antisense oligonucleotide trabedersen) into the tumor, using convection-enhanced delivery (CED) via a single intratumoral catheter connected to a portable pump, has already been explored (Jaschinski, Rothhammer et al. 2011). Notably, bearing in mind the enhanced colloidal stability of lipopolyplexes over their polyplex counterparts, this also becomes feasible here.

We agree with the reviewer that this deserved further clarification and discussion, and we have made appropriate additions to the text (Discussion section).

3.     There is less discussion relating immune study and signal pathway. What is the significance of those gene regulation and immune suppression? 

Answer: Since we did not find any significant differences in the number of tumor-infiltrating immune cells, we had decided to not emphasize this aspect of the work in the first version of the manuscript. We agree that we could have discussed the implications of these results in more detail and have now followed this advice of the reviewer. As for the signaling pathways we assume the reviewer is referring to the Stat3 targets genes (Mmp2, Mmp9 and Snai1) that we analyzed after microdissecting the tumor center. We have used these three genes as surrogate markers for active Stat3 signaling since all of them are known target genes of Stat3 (Luwor, Baradaran et al. 2013, Wendt, Balanis et al. 2014) and because they regulate important aspects of GBM aggressiveness like the highly migrative (Snai1 (Kudo-Saito, Shirako et al. 2009)) and invasive phenotype (Mmp2 and Mmp9 (Westermarck and Kahari 1999)) and immunosuppression (Snai1). The facts that we see a tendency towards decreased expression of Mmp9 and Snai1 provide mechanistic clues related to increased survival. Hence, a decrease in Mmp9 expression might reduce the migratory capacity of the tumor cells and therefore might decrease single cell infiltration into the surrounding brain parenchyma. This is further emphasized by Snai1-repression which is a master regulator of EMT. Additionally, Snai1 has been described to facilitate the immunosuppressive phenotype of GBM and LPP siStat3-mediated Snai1-depletion might therefore alleviate this phenotype, making the tumor more accessible to immune cells and/or immunotherapy. We have now carefully addressed these hypotheses in more detail in the manuscript (lines 357 to 364).

References

Ewe, A., O. Panchal, S. R. Pinnapireddy, U. Bakowsky, S. Przybylski, A. Temme and A. Aigner (2017). "Liposome-polyethylenimine complexes (DPPC-PEI lipopolyplexes) for therapeutic siRNA delivery in vivo." Nanomedicine 13(1): 209-218.

Ewe, A., A. Schaper, S. Barnert, R. Schubert, A. Temme, U. Bakowsky and A. Aigner (2014). "Storage stability of optimal liposome-polyethylenimine complexes (lipopolyplexes) for DNA or siRNA delivery." Acta Biomater 10(6): 2663-2673.

Heyes, J., L. Palmer, K. Chan, C. Giesbrecht, L. Jeffs and I. MacLachlan (2007). "Lipid encapsulation enables the effective systemic delivery of polyplex plasmid DNA." Mol Ther 15(4): 713-720.

Jaschinski, F., T. Rothhammer, P. Jachimczak, C. Seitz, A. Schneider and K. H. Schlingensiepen (2011). "The antisense oligonucleotide trabedersen (AP 12009) for the targeted inhibition of TGF-beta2." Curr Pharm Biotechnol 12(12): 2203-2213.

Ko, Y. T., A. Kale, W. C. Hartner, B. Papahadjopoulos-Sternberg and V. P. Torchilin (2009). "Self-assembling micelle-like nanoparticles based on phospholipid-polyethyleneimine conjugates for systemic gene delivery." J Control Release 133(2): 132-138.

Kudo-Saito, C., H. Shirako, T. Takeuchi and Y. Kawakami (2009). "Cancer metastasis is accelerated through immunosuppression during Snail-induced EMT of cancer cells." Cancer Cell 15(3): 195-206.

Luwor, R. B., B. Baradaran, L. E. Taylor, J. Iaria, T. V. Nheu, N. Amiry, C. M. Hovens, B. Wang, A. H. Kaye and H. J. Zhu (2013). "Targeting Stat3 and Smad7 to restore TGF-beta cytostatic regulation of tumor cells in vitro and in vivo." Oncogene 32(19): 2433-2441.

Wendt, M. K., N. Balanis, C. R. Carlin and W. P. Schiemann (2014). "STAT3 and epithelial-mesenchymal transitions in carcinomas." JAKSTAT 3(1): e28975.

Westermarck, J. and V. M. Kahari (1999). "Regulation of matrix metalloproteinase expression in tumor invasion." FASEB J 13(8): 781-792.

Reviewer 3 Report

cancers-422130

Dear Editor,

The authors assessed a new gene therapy efficacy by intra-cranially injected siRNA-STAT3, complexed into a lipopolyplexes (LPP), in mice implanted at the striatum level with murine glioma cells. Before the in vivo experiments, the authors in vitro evaluated the anticancer potential of the LPP-siRNA-STAT3.

This work is interesting because it addresses different main problems to treat GBM: a specific delivery, a specific target of glioma cells and avoiding degradation of the drug after the delivery.

The authors try to reach all these points, which probably constituted a promising future, but the manuscript suffers from some weaknesses, mainly about experimental approach. They are listed below.

Introduction section

It is well constructed but some information are missing about

-          Phosphorylation status of Stat3, particularly the form pY705Stat3, used in different experiments.

-          The interest of studying Iba1, MMP2, MMP9 and snail genes.

Material and methods section

The nanoparticle characterization is out of my expertise area then should be reviewed by another expert.

The number of repetition performed by experiments is not informed.

§4.2. The authors should explain why they used two different human GBM cell lines. What about their differences?

The § 4.6 needs to be entirely re written. It is too tufted and information are difficult to find. Nature of cells treated, time of treatment,… etc., must be specified.

A separate § concerning 1/cell transfection (which cells?), 2/Luciferase activity 3/EGFP KO and 4/proliferation assays should be produced.

The § 4.7 needs explanations about the ways of treating the data. Indeed, the authors used the DDCt method, then the control should be fixed at 1AU, which is not the case in Fig.2b and Fig.6b. In this last fig, as the authors utilized two normalization genes (HPRT and TBP), they should be both exploited to generate a more robust result in one graph.

The § 4.10 needs revision: how many mice were implanted, how many mice per group and above all, what about toxicity of siRNA treatment? Weights of mice should be given as well as hepatic toxicity information.

Application of 0.5µg siRNA must be discussed: why this quantity, whereas it was used 0.8µg siRNA on 30000 cells?

Cervical dislocation sacrifice here must be discussed as this king of method should be completely avoided to prevent brain damage at the moment of the dislocation.

Results section

Globally, the result part is quite difficult to read as some information are missing (see M&M section) and because the resolution of all the figures is too low combined with a too small police. Moreover, the legends should give much more material to enhance the comprehension of the experiments.

For example, in the Fig.1e, the reader has any information about the utility of Luc3 vs Luc 2, the cells used (we suppose Tu2449?).

§2.2: the cell cycle results should be added in this section as they are directly linked to proliferation ones. Why the authors did not give U87 and Mz18 cell cycle data?

To be noted that all the experiments, particularly cell proliferation experiments would have been more relevant on Tu and U87 spheroids, which mimics the tumor, facilitating the link with in vivo results.

§ 2.3: Why the authors did not design an untreated group?

§2.4/2.5: the main issue here is the stable expression of stat3, after the treatment. Normally, the gene therapy aims at decreasing this expression, then, how the authors can explain this result? It is also deceiving that Ki67 expression did not decrease.

The authors tried to answer to this problem by measuring Stat3 transcript expression, which seemed slightly decreased, despite some inconsistency about data analyses (specified above). Moreover, line 220-221, authors stipulated that the number of samples treated depended on the nature of the house keeping gene used. Please explain.

Consequently, Fig.6 a and b are not receivable, especially since statistical analyses are not informed. Then, the conclusion about the model and the in vivo results is not acceptable, and the last paragraph before the discussion must be moderated.

If possible this last experiment should be performed again by analyzing Stat3 protein level by WB, to be sure it is less expressed into the center of the tumor.

Discussion section

All this section should be moderated to a better fitting with the results obtained. For example, lines 271-272.

The authors should discuss the new methods used to administer drugs in situ, such as multiport catheter.

The authors should discuss the toxicity associated with gene therapy by siRNA and the specificity of siRNA sequence.

Author Response

Dear Editor,

The authors assessed a new gene therapy efficacy by intra-cranially injected siRNA-STAT3, complexed into a lipopolyplexes (LPP), in mice implanted at the striatum level with murine glioma cells. Before the in vivo experiments, the authors in vitro evaluated the anticancer potential of the LPP-siRNA-STAT3.

This work is interesting because it addresses different main problems to treat GBM: a specific delivery, a specific target of glioma cells and avoiding degradation of the drug after the delivery.

The authors try to reach all these points, which probably constituted a promising future, but the manuscript suffers from some weaknesses, mainly about experimental approach. They are listed below.

Answer: We appreciate the Reviewer’s overall positive assessment of our work. His / her specific points are addressed in the revised manuscript as detailed below:

Introduction section

It is well constructed but some information are missing about

-          Phosphorylation status of Stat3, particularly the form pY705Stat3, used in different experiments.

Answer: We assume the Reviewer asks for clarification about the relevance of pStat3 and the reason why we specifically analyzed pY705Stat3. To regulate gene expression, Stat3 needs to be phosphorylated by many upstream kinases including Janus Kinases (JAK2) (Avalle, Camporeale et al. 2017). Hence pStat3 is a good marker for active Stat3-signaling (Brantley and Benveniste 2008). We focused on the Y705-phosphorylation, because it has been shown that this canonical Stat3 phosphorylation is the better readout for Stat3 activation, rather than the S737-phosphorylation, which is needed for maximal activation of Stat3 in glioma, but is dependent on initial Y705-phosphorylation (Ganguly, Fan et al. 2018). We have now added these information in the introduction section (lines 60 to 64).

-          The interest of studying Iba1, MMP2, MMP9 and snail genes.

Answer: We studied Iba1 as a marker for microglia (Hambardzumyan, Gutmann et al. 2016), because 1) about one third of the cells within the tumor of our GBM-model are microglia, which is consistent with findings in the human situation (Hambardzumyan, Gutmann et al. 2016) and it is important to check whether this changes after treatment because 2) Stat3 is known to be important for microglia activation (Planas, Soriano et al. 1996, Li and Graeber 2012), which 3) could alleviate the pro-tumorigenic microglia phenotype that is usually found in GBM (Li and Graeber 2012).

As for the analysis of Mmp2, Mmp9 and Snai1: we analyzed these genes for three reasons: 1) Those genes are known to be regulated by Stat3 signaling (Luwor, Baradaran et al. 2013, Wendt, Balanis et al. 2014) and can therefore be used as a surrogate for Stat3-activation (especially in cases where a Western Blot is not possible, like for the LCM-approach). 2) All three genes are important for GBM aggressiveness, since Mmps mediate the infiltrative phenotype via modification of the ECM (Westermarck and Kahari 1999) and Snai1 is a master regulator of epithelial-mesenchymal-transition and mediates an immunosuppressive phenotype (Kudo-Saito, Shirako et al. 2009). 3) We could previously show that stable depletion of Stat3 using lentivirally delivered shRNA that all genes are depleted as well in the same cell system (Tu2449, (Priester, Copanaki et al. 2013)).

We thank the Reviewer for bringing this to our attention and have now included these further details in the introduction section.

Material and methods section

The nanoparticle characterization is out of my expertise area then should be reviewed by another expert.

The number of repetition performed by experiments is not informed.

Answer: Thank you for bringing this to our attention. We have now provided repetition numbers.

§4.2. The authors should explain why they used two different human GBM cell lines. What about their differences?

Answer: We have used two different cell lines in order the address the known heterogeneity of GBM (Friedmann-Morvinski 2014) and because we wanted to be certain that we observe generalizable effects and not cell-line specific artifacts. U87 cells are known to be particularly dependent on STAT3 (Dasgupta, Raychaudhuri et al. 2009) and are therefore a good model to study STAT3 depletion. Mz18 were chosen as second human GBM cell line, which we also previously used to analyze JAK2/STAT3-inhibition, but which expresses rather moderate levels of pSTAT3 and might therefore be not as dependent on STAT3 asU87 (Senft, Priester et al. 2011).

The § 4.6 needs to be entirely re written. It is too tufted and information are difficult to find. Nature of cells treated, time of treatment,… etc., must be specified.

A separate § concerning 1/cell transfection (which cells?), 2/Luciferase activity 3/EGFP KO and 4/proliferation assays should be produced.

Answer: Following the Reviewer’s suggestion, we have reorganized and fine-structured this part of the Materials and Methods section. More specifically, 4.9 Cell transfection, 4.10 Determination of knockdown efficacies, 4.11 Cell proliferation and viability assays, 4.12 cell cycle analysis and 4.13 Spheroid Assay now are provided as separate subsections.

In all protocols, the cell lines are now specified. Where missing, this information has also been added in the Results section. In all protocols, the time of treatment / time point of measurement or termination of the experiment is given. Additionally, we have now re-structured the Method-Section in order to provide a more coherent reading experience.

The § 4.7 needs explanations about the ways of treating the data. Indeed, the authors used the DDCt method, then the control should be fixed at 1AU, which is not the case in Fig.2b and Fig.6b. In this last fig, as the authors utilized two normalization genes (HPRT and TBP), they should be both exploited to generate a more robust result in one graph.

Answer: Thank you for pointing this out. We forgot to mention that the “+”-symbol in the Box-Whisker-Plots denotes the mean, whereas the horizontal line depicts the median. We have now added these informations into the figure legends and §4.17. Accordingly, it should now be conceivable that the control-conditions are fixed at 1.

As for Fig 2b and c: We normalized all conditions to siCtrl and not the untreated cells, since this is the correct control treatment. The Reviewer likely assumed that we normalized to untreated cells. We have now further clarified these discrepancies.

As for Fig. 6b (now 7b): In total we prepared LCM-slides for 6 samples per group. But due to the low amounts of RNA that could be isolated (and/or quality therefore) we could only obtain Ct-values for Hprt for three samples of each group. For Tbp we could obtain data from one additional sample per group. Hence, we decided to present the data independently in order to not lose the data from one animal per group. We have now clarified this in more detail in the result section.

The § 4.10 needs revision: how many mice were implanted, how many mice per group and above all, what about toxicity of siRNA treatment? Weights of mice should be given as well as hepatic toxicity information.

Answer: We have now added the requested information. We have shown previously that no hepatotoxicity was observed even after repeated systemic administration of our LPP over > 2 weeks (Ewe, Panchal et al. 2017). Since even this previous mode of injection, favoring hepatic delivery, did not yield toxic effects, no hepatotoxicity can be expected from local application of considerably smaller amounts (see below). We thus did not repeat hepatotoxicity experiments in this study. Nonetheless, the animals were monitored daily (sometimes twice a day) and we did not observe any detectable signs of toxicity.

The average mouse weight at the time point of the experiment was ~ 25 g. This information has been added to the manuscript.

Application of 0.5µg siRNA must be discussed: why this quantity, whereas it was used 0.8µg siRNA on 30000 cells?

Answer: Due to limitations regarding the total lipopolyplex volume we were allowed to inject, a lipopolyplex amount equivalent to 0.5 µg siRNA was the maximum possible in this experimental setting. One should also bear in mind, however, that in previous in vivo experiments we had used LPP containing only 10 µg siRNA for **systemic** injection (Ewe, Panchal et al. 2017). Given the ratio body weight / brain weight (of course, incorrectly assuming a homogenous biodistribution within the whole body after systemic injection), this means that the amounts here used for local application were even higher than in the previous in vivo studies.

To make this point (limitations regarding the total lipopolyplex volume) clearer, we have made appropriate additions to the text (Materials and Methods, Discussion).

Cervical dislocation sacrifice here must be discussed as this king of method should be completely avoided to prevent brain damage at the moment of the dislocation.

Answer: It is true that, if done carelessly, cervical dislocation can lead to brain damage. However we did this on mice that received a lethal injection of an anesthetic and therefore very little force was required. Besides, even if some damage might have been afflicted this would likely be restricted to brain stem, the pons or the cerebellum, anatomic structures that are not important for this study. We regret that we had forgotten to mention the lethal injection in our first submission and have now included this information.

Results section

Globally, the result part is quite difficult to read as some information are missing (see M&M section) and because the resolution of all the figures is too low combined with a too small police. Moreover, the legends should give much more material to enhance the comprehension of the experiments.

Answer: We regret that the Reviewer deemed our figures of low quality/resolution. We can only assume that this was due to the processing during the upload and/or the embedding in MS Word, because we prepared all figures from high resolution and/or vector graphics and exported them as 600 dpi figures. We will of course provide the high resolution figures as separate files should this issue persist. As for the figure legends: we have now added more information and are confident that the readability is now vastly improved.

For example, in the Fig.1e, the reader has any information about the utility of Luc3 vs Luc 2, the cells used (we suppose Tu2449?).

Answer: Thank you for pointing this out. As also noted by Rev. 1 did we forget to add some information at this point. We regret this mistake and have now added the missing information.

Additionally, most information is given in the text (Materials and Methods, Results) rather than in the Figure legends. Following the Reviewer’s suggestion, however, we have now enhanced the figure legends. We revised the Figure legends and added missing information for clarification. By making appropriate additions, more information is now given in the Figure legends.

The Reviewer is correct that information on Luc2 vs. Luc3 was entirely missing and that the difference between siRNAs Luc2 and Luc3 had not been properly explained. These siRNAs target the Promega Luciferase vectors pGL2 and pGL3, respectively. In the reporter cell lines used in this study, pGL3 is stably integrated; thus, siRNA Luc3 is the specific siRNA while Luc2 served as negative control (Fig. 1g). In Fig. 1f siRNA EGFP is the specific siRNA, with again siLuc2 serving as negative control. In Fig. 1e, both siRNAs (Luc2 and Luc3) are shown simply to make the point that (absence of) toxicity is independent of the siRNA sequences, i.e., identical results are obtained for both siRNAs.

§2.2: the cell cycle results should be added in this section as they are directly linked to proliferation ones. Why the authors did not give U87 and Mz18 cell cycle data?

Answer: We have now moved the entire figure (previously Figure S3) into the main text and have labeled it as new Figure 3. In the light of the subsequent in vivo data obtained in our syngeneic mouse model, we figured that we should rather focus on the murine cell line. However, following the Reviewer’s suggestion, we have added U87 and Mz18 cell cycle data to the revised manuscript (new Suppl. Fig. S3 b, c). Please note that these data were obtained in the presence of nocodazole. Thus, we have also added a description of the procedure to the Suppl. Materials and Methods section as well as an appropriate explanation to the Figure legend.

To be noted that all the experiments, particularly cell proliferation experiments would have been more relevant on Tu and U87 spheroids, which mimics the tumor, facilitating the link with in vivo results.

Answer: We did perform cell proliferation in Tu2449 cells (Fig. 2 g and h). In fact, we even compared proliferation changes after siRNA transfection using a conventional transfection reagent (Interferin, Fig. 2g) and also tested LPP in vitro (Fig. 2h) to provide functional proof of LPP-mediated siRNA-delivery. We have now further emphasized these points in the result section and apologize for not describing it clearly enough.

As for the experiments in spheroids: We did analyze changes in sphere growth after LPP siStat3 using Tu2449 spheroids (previously Fig S3 b and C; now Fig. 3 b and c) and could successfully show growth inhibition. We only performed these experiments using Tu2449 because 1) we wanted to proof functional validity of LPP siRNA-delivery in a more complex model and 2) the Tu2449 cell line is the line used for the in vivo experiments and should therefore be analyzed in more detail compared to the human GBM cell lines. Considering that we perform siRNA-mediated Stat3-depletion we believe that these experiments are the most relevant, because the siRNAs aren’t interchangeable between human und murine cancers. Hence, by using LPP siStat3 for Tu2449 Proliferation and sphere growth we provide functional proof of 1) siRNA efficiency and 2) LPP-mediated delivery of siRNA.

§ 2.3: Why the authors did not design an untreated group?

Answer: We did not design an untreated group because the applied mouse model is very well established (Pohl, Wick et al. 1999) and untreated tumor-bearing mice can hardly be compared to control-treated mice since they will not experience the same level of stress as treated mice. Since the treatment regimen is rather stressful (in total 6 intracranial injections (1 for tumor implantation; 5 for treatment); all under deep anesthesia) we reasoned that an untreated group is a very unreliable control. The nanoparticle treatment may in theory be well associated with non-specific effects from the nanoparticles, or their polymeric, lipid or RNA contents. Thus, the by far most important negative control is precisely the same treatment with the same formulation, except for using a non-specific, irrelevant siRNA. It is one beauty of the siRNA field that the design of negative controls is very straight forward by providing all components and only altering the siRNA sequence. We therefore focused on this negative control group.

§2.4/2.5: the main issue here is the stable expression of stat3, after the treatment. Normally, the gene therapy aims at decreasing this expression, then, how the authors can explain this result? It is also deceiving that Ki67 expression did not decrease.

Answer: We agree with the Reviewer that a robust decrease in Stat3 expression would have been expected, especially considering the pronounced increase in survival. We hypothesize that 1) intratumural diffusion/distribution of the LPP complexes is somewhat limited and should be addressed in due detail in future studies, maybe even in a comparative manner in different cancers. We propose that only a subset of tumor cells is reached, while the rest can retain high Stat3 expression. 2) We chose a rather aggressive GBM model that usually kills the animals after 21 days and leads to very large tumors, as previously shown (Priester, Copanaki et al. 2013). Hence, we believe that those cells that are targeted by LPPsiStat3 will have a growth disadvantage (thus, explaining the decrease in tumor size), but those will very quickly be overgrown by other cells. This in turn might also explain why we only observed a tendency towards Stat3 depletion. We regret if we had not discussed this in due detail and have now extended the discussion section concerning these discrepancies.

The authors tried to answer to this problem by measuring Stat3 transcript expression, which seemed slightly decreased, despite some inconsistency about data analyses (specified above). Moreover, line 220-221, authors stipulated that the number of samples treated depended on the nature of the house keeping gene used. Please explain.

Answer: Obviously, the number of samples analyzed did not depend on the housekeeping gene and we regret if our writing was not specific enough. We performed laser-capture-microdissection (LCM) of 6 animals per group, but did not only obtain quantifiable data for 4 animals per group when measuring Tbp expression (previously Fig. 6b, now 7b) and 3 animals per group for Hprt (previously Fig. 6c, now 7c). Therefore we present the data normalized to both housekeepers separately. Of course the three samples per group normalized to Hprt could also been normalized to Tbp. We could just analyze one additional sample if normalized to Tbp. We have now added further explanations about these matters and are confident that the clarity of the chosen approach is now improved.

Consequently, Fig.6 a and b are not receivable, especially since statistical analyses are not informed.

Answer: We regret that we missed to provide details about statistical analyses in the figure legends and have now added this information. As already stated in the main text was the only significant difference a reduction of Mmp9 expression if normalized to Tbp. We also stated that this is likely due to the low sample sizes. Nonetheless do we believe that the data obtained from the LCM-approach provides important insights that can explain some of the previously discussed discrepancies.

Then, the conclusion about the model and the in vivo results is not acceptable, and the last paragraph before the discussion must be moderated.

Answer: We agree that our initial statement were rather strong and not fully supported by the data. We have now re-written this paragraph with more caution. The text reads as follows (lines 300 to 305):

These results support our previously proposed hypothesis that LPP siStat3 can specifically target cancer cells to limit tumor growth in vivo. Despite the robust effects of LPP-formulated siStat3 on tumor growth, our results also indicate that only a small fraction of tumor cells can be reached using the current formulations and mode of application. These observations suggest that improvement of the bioavailability may allow further enhancement of therapeutic efficacies of this approach in vivo.

If possible this last experiment should be performed again by analyzing Stat3 protein level by WB, to be sure it is less expressed into the center of the tumor.

Answer: Unfortunately, this is not possible. During the LCM-approach we used up most of the material. In fact, we initially reasoned that LCM followed by RNA-Isolation using dedicated Kits and cDNA-Synthesis using state-of-the-art Reverse Transcriptases (SS IV Vilo; life technologies) would likely offer the most output in terms of RNA quantitiy and possible transcripts. Additionally, we routinely use Taqman-probes for qRT-PCR which usually have smaller transcripts and are therefore suitable for amplification of complex, potentially degraded, samples.

Discussion section

All this section should be moderated to a better fitting with the results obtained. For example, lines 271-272.

The authors should discuss the new methods used to administer drugs in situ, such as multiport catheter.

The authors should discuss the toxicity associated with gene therapy by siRNA and the specificity of siRNA sequence.

Answer: We agree with the Reviewer that we should have been more cautious with our statements and we have carefully edited the discussion. We also extended the discussion about novel delivery methods and provide more details about the siRNA. We thank the Reviewer for these suggestions.

References

Avalle, L., A. Camporeale, A. Camperi and V. Poli (2017). "STAT3 in cancer: A double edged sword." Cytokine 98: 42-50.

Brantley, E. C. and E. N. Benveniste (2008). "Signal transducer and activator of transcription-3: a molecular hub for signaling pathways in gliomas." Molecular cancer research : MCR 6(5): 675-684.

Dasgupta, A., B. Raychaudhuri, T. Haqqi, R. Prayson, E. G. Van Meir, M. Vogelbaum and S. J. Haque (2009). "Stat3 activation is required for the growth of U87 cell-derived tumours in mice." Eur J Cancer 45(4): 677-684.

Ewe, A., O. Panchal, S. R. Pinnapireddy, U. Bakowsky, S. Przybylski, A. Temme and A. Aigner (2017). "Liposome-polyethylenimine complexes (DPPC-PEI lipopolyplexes) for therapeutic siRNA delivery in vivo." Nanomedicine 13(1): 209-218.

Friedmann-Morvinski, D. (2014). "Glioblastoma heterogeneity and cancer cell plasticity." Crit Rev Oncog 19(5): 327-336.

Ganguly, D., M. Fan, C. H. Yang, B. Zbytek, D. Finkelstein, M. F. Roussel and L. M. Pfeffer (2018). "The critical role that STAT3 plays in glioma-initiating cells: STAT3 addiction in glioma." Oncotarget 9(31): 22095-22112.

Hambardzumyan, D., D. H. Gutmann and H. Kettenmann (2016). "The role of microglia and macrophages in glioma maintenance and progression." Nat Neurosci 19(1): 20-27.

Kudo-Saito, C., H. Shirako, T. Takeuchi and Y. Kawakami (2009). "Cancer metastasis is accelerated through immunosuppression during Snail-induced EMT of cancer cells." Cancer Cell 15(3): 195-206.

Li, W. and M. B. Graeber (2012). "The molecular profile of microglia under the influence of glioma." Neuro Oncol 14(8): 958-978.

Luwor, R. B., B. Baradaran, L. E. Taylor, J. Iaria, T. V. Nheu, N. Amiry, C. M. Hovens, B. Wang, A. H. Kaye and H. J. Zhu (2013). "Targeting Stat3 and Smad7 to restore TGF-beta cytostatic regulation of tumor cells in vitro and in vivo." Oncogene 32(19): 2433-2441.

Planas, A. M., M. A. Soriano, M. Berruezo, C. Justicia, A. Estrada, S. Pitarch and I. Ferrer (1996). "Induction of Stat3, a signal transducer and transcription factor, in reactive microglia following transient focal cerebral ischaemia." Eur J Neurosci 8(12): 2612-2618.

Pohl, U., W. Wick, J. Weissenberger, J. P. Steinbach, J. Dichgans, A. Aguzzi and M. Weller (1999). "Characterization of Tu-2449, a glioma cell line derived from a spontaneous tumor in GFAP-v-src-transgenic mice: comparison with established murine glioma cell lines." Int J Oncol 15(4): 829-834.

Priester, M., E. Copanaki, V. Vafaizadeh, S. Hensel, C. Bernreuther, M. Glatzel, V. Seifert, B. Groner, D. Kogel and J. Weissenberger (2013). "STAT3 silencing inhibits glioma single cell infiltration and tumor growth." Neuro-oncology 15(7): 840-852.

Senft, C., M. Priester, M. Polacin, K. Schroder, V. Seifert, D. Kogel and J. Weissenberger (2011). "Inhibition of the JAK-2/STAT3 signaling pathway impedes the migratory and invasive potential of human glioblastoma cells." Journal of neuro-oncology 101(3): 393-403.

Wendt, M. K., N. Balanis, C. R. Carlin and W. P. Schiemann (2014). "STAT3 and epithelial-mesenchymal transitions in carcinomas." JAKSTAT 3(1): e28975.

Westermarck, J. and V. M. Kahari (1999). "Regulation of matrix metalloproteinase expression in tumor invasion." FASEB J 13(8): 781-792.

Round  2

Reviewer 1 Report

The authors have considerably improved the manuscript, although no changes were made to the in vivo part. I understand that limitation in terms of tissue availability exist, and the authors are not willing to carry out additional, time consuming in vivo experiments. Nevertheless, I still do not fully agree with the authors, that only limited tissue distribution of the siRNA formulation restricts efficient target knockdown. There are several other factors limiting in vivo efficiency, including stability of the formulation in the tissue prior to cell internalization (ECM interaction, etc) and of course endosomal release. In addition, the low amount of siRNA applied (limited to the low concentration of the formulation) is another bottleneck. Especially in view of the fact that additional in vivo data are shown, and limitations in terms of biodistribution have not been proven by the authors (and there are possibilities to use stable siRNA-fluorophore for biodistribution studies, which are not prone to degradation within the first few hours after injection), this has to be at least thoroughly discussed in the modified manuscript (including adequate references on this topic, i.e. intracranial siRNA delivery in brain cancer models).

Author Response

The authors have considerably improved the manuscript, although no changes were made to the in vivo part. I understand that limitation in terms of tissue availability exist, and the authors are not willing to carry out additional, time consuming in vivo experiments. Nevertheless, I still do not fully agree with the authors, that only limited tissue distribution of the siRNA formulation restricts efficient target knockdown. There are several other factors limiting in vivo efficiency, including stability of the formulation in the tissue prior to cell internalization (ECM interaction, etc) and of course endosomal release. In addition, the low amount of siRNA applied (limited to the low concentration of the formulation) is another bottleneck. Especially in view of the fact that additional in vivo data are shown, and limitations in terms of biodistribution have not been proven by the authors (and there are possibilities to use stable siRNA-fluorophore for biodistribution studies, which are not prone to degradation within the first few hours after injection), this has to be at least thoroughly discussed in the modified manuscript (including adequate references on this topic, i.e. intracranial siRNA delivery in brain cancer models).

Answer2: We would like to thank the reviewer for the suggestions that led to the improvements of the manuscript. As for the additional comments the reviewer is of course correct. The limited knockdown efficiency likely is a multifactorial issue and to state biodistribution as the sole contributing factor was an oversimplification. We have now further added the mentioned points and also discussed potential ways to track siRNA biodistribution in due detail.

We also appreciate the Reviewer’s comments highlighting critical aspects of nanoparticle-mediated siRNA delivery which we regard as very important. Therefore, already in our previous publications, we have rather extensively characterized LPP stabilities. In this context, two parameters are important: on the one hand, we could show that, also in the context of biological media, the lipopolyplexes are stable against decomposition, thus efficiently protecting the siRNA even against RNAse directly added to the assay. Furthermore, one must also keep in mind that nanoparticles may be prone to aggregation, thus leading to impaired cellular uptake due to colloidal stability. While this was observed for some of our PEI-based polyplexes, the lipopolyplexes employed here were found to be substantially more stable (Ewe et al. 2014). These findings were also supported by our previous in vivo therapy studies based on systemic application for reaching subcutaneous tumor xenografts, and prompted us to now proceed towards a more specialized and sophisticated application of our LPP, i.e., for the treatment of glioma.

The Reviewer is also correct in pointing out that in other publications larger siRNA amounts have been used. In this paper, we were limited amount-wise by the maximum tolerable injection volume which, due to limitations regarding the LPP concentration, determined the maximum dosage we could administer to the mice. We are currently looking into formulations which, by requiring lesser polymer amounts, will allow to generate more highly concentrated nanoparticles, thus containing more siRNA. We appreciate the Reviewer’s observation that comparably small siRNA amounts have been used so far, offering options for improved efficacy.

Finally, we have looked rather extensively into the issue of biodistribution. With regard to systemic delivery followed by biodistribution throughout the body, these data have been published previously; see Ewe et al, Nanomedicine 2017. In the context of this manuscript, however, which is not based on systemic administration, the local penetration of the LPP into the tissue should be considered as more relevant and must be expected to be rate-limiting. Being aware of the limitation of 2D cell culture to address this important point, we have explored organotypic tissue slice cultures of intact tumor (xenograft) material with regard to nanoparticle tissue penetration. Indeed, when using fluorophore-labeled siRNAs for microscopic evaluation of tissue penetrance, we did observe LPP to diffuse into the tissue – to a certain extent. LPP tissue penetration was found to be better than for polyplexes, reflecting lesser impairment of nanoparticle diffusion with reduced surface charge (zeta potential; (Merz et al. 2017)). However, in another in vivo system (non-tumorous mouse brain in an alpha-synuclein mouse model), even our polyplexes were found to distribute across the CNS down to the lumbar spinal cord after a single intracerebroventricular infusion, thus emphasizing a broader distribution sufficient for exerting biological effects (Helmschrodt et al. 2017).

Following the Reviewer’s comments, we have made appropriate additions to the manuscript and discussed other approaches for intracranial siRNA delivery.

Submission Date

21 December 2018

Date of this review

11 Feb 2019 08:52:14

References Revision 2:

Ewe, A., A. Schaper, S. Barnert, R. Schubert, A. Temme, U. Bakowsky and A. Aigner (2014). "Storage stability of optimal liposome-polyethylenimine complexes (lipopolyplexes) for DNA or siRNA delivery." Acta Biomater 10(6): 2663-2673.

Helmschrodt, C., S. Hobel, S. Schoniger, A. Bauer, J. Bonicelli, M. Gringmuth, . . . F. Richter (2017). "Polyethylenimine Nanoparticle-Mediated siRNA Delivery to Reduce alpha-Synuclein Expression in a Model of Parkinson's Disease." Mol Ther Nucleic Acids 9: 57-68.

Merz, L., S. Hobel, S. Kallendrusch, A. Ewe, I. Bechmann, H. Franke, . . . A. Aigner (2017). "Tumor tissue slice cultures as a platform for analyzing tissue-penetration and biological activities of nanoparticles." Eur J Pharm Biopharm 112: 45-50.

Reviewer 2 Report

The authors have considerably improved the manuscript. However, the stability of siRNA formulation, the low concentration of siRNA, and its biodistribution in vivo have to be discussed and studied. 

Author Response

The authors have considerably improved the manuscript. However, the stability of siRNA formulation, the low concentration of siRNA, and its biodistribution in vivo have to be discussed and studied. 

Answer2: We would like to thank the reviewer for acknowledging our changes.

We appreciate the Reviewer’s comments highlighting critical aspects of nanoparticle-mediated siRNA delivery which we regard as very important. Therefore, already in our previous publications, we have rather extensively characterized LPP stabilities. In this context, two parameters are important: on the one hand, we could show that, also in the context of biological media, the lipopolyplexes are stable against decomposition, thus efficiently protecting the siRNA even against RNAse directly added to the assay. Furthermore, one must also keep in mind that nanoparticles may be prone to aggregation, thus leading to impaired cellular uptake due to colloidal stability. While this was observed for some of our PEI-based polyplexes, the lipopolyplexes employed here were found to be substantially more stable (Ewe et al. 2014). These findings were also supported by our previous in vivo therapy studies based on systemic application for reaching subcutaneous tumor xenografts, and prompted us to now proceed towards a more specialized and sophisticated application of our LPP, i.e., for the treatment of glioma.

The Reviewer is also correct in pointing out that in other publications larger siRNA amounts have been used. In this paper, we were limited amount-wise by the maximum tolerable injection volume which, due to limitations regarding the LPP concentration, determined the maximum dosage we could administer to the mice. We are currently looking into formulations which, by requiring lesser polymer amounts, will allow generating more highly concentrated nanoparticles, thus containing more siRNA. We appreciate the Reviewer’s observation that comparably small siRNA amounts have been used so far, offering options for improved efficacy.

Finally, we have looked rather extensively into the issue of biodistribution. With regard to systemic delivery followed by biodistribution throughout the body, these data have been published previously; see Ewe et al, Nanomedicine 2017. In the context of this manuscript, however, which is not based on systemic administration, the local penetration of the LPP into the tissue should be considered as more relevant and must be expected to be rate-limiting. Being aware of the limitation of 2D cell culture to address this important point, we have explored organotypic tissue slice cultures of intact tumor (xenograft) material with regard to nanoparticle tissue penetration. Indeed, when using fluorophore-labeled siRNAs for microscopic evaluation of tissue penetrance, we did observe LPP to diffuse into the tissue – to a certain extent. LPP tissue penetration was found to be better than for polyplexes, reflecting lesser impairment of nanoparticle diffusion with reduced surface charge (zeta potential; (Merz et al. 2017)). However, in another in vivo system (non-tumorous mouse brain in an alpha-synuclein mouse model), even our polyplexes were found to distribute across the CNS down to the lumbar spinal cord after a single intracerebroventricular infusion, thus emphasizing a broader distribution sufficient for exerting biological effects (Helmschrodt et al. 2017).

Following the Reviewer’s comments, we have made appropriate additions to the manuscript.

Submission Date

21 December 2018

Date of this review

11 Feb 2019 11:51:06

References Revision 2:

Ewe, A., A. Schaper, S. Barnert, R. Schubert, A. Temme, U. Bakowsky and A. Aigner (2014). "Storage stability of optimal liposome-polyethylenimine complexes (lipopolyplexes) for DNA or siRNA delivery." Acta Biomater 10(6): 2663-2673.

Helmschrodt, C., S. Hobel, S. Schoniger, A. Bauer, J. Bonicelli, M. Gringmuth, . . . F. Richter (2017). "Polyethylenimine Nanoparticle-Mediated siRNA Delivery to Reduce alpha-Synuclein Expression in a Model of Parkinson's Disease." Mol Ther Nucleic Acids 9: 57-68.

Merz, L., S. Hobel, S. Kallendrusch, A. Ewe, I. Bechmann, H. Franke, . . . A. Aigner (2017). "Tumor tissue slice cultures as a platform for analyzing tissue-penetration and biological activities of nanoparticles." Eur J Pharm Biopharm 112: 45-50.

Reviewer 3 Report

Second round

Dear Editor,

The authors assessed a new gene therapy efficacy by intra-cranially injected siRNA-STAT3, complexed into a lipopolyplexes (LPP), in mice implanted at the striatum level with murine glioma cells. Before the in vivo experiments, the authors in vitro evaluated the anticancer potential of the LPP-siRNA-STAT3.

This work is interesting because it addresses different main problems to treat GBM: a specific delivery, a specific target of glioma cells and avoiding degradation of the drug after the delivery.

The authors try to reach all these points, which probably constituted a promising future, but the manuscript suffers from some weaknesses, mainly about experimental approach. They are listed below.

Answer: We appreciate the Reviewer’s overall positive assessment of our work. His / her specific points are addressed in the revised manuscript as detailed below:

Introduction section

It is well constructed but some information are missing about

-          Phosphorylation status of Stat3, particularly the form pY705Stat3, used in different experiments.

Answer: We assume the Reviewer asks for clarification about the relevance of pStat3 and the reason why we specifically analyzed pY705Stat3. To regulate gene expression, Stat3 needs to be phosphorylated by many upstream kinases including Janus Kinases (JAK2) (Avalle, Camporeale et al. 2017). Hence pStat3 is a good marker for active Stat3-signaling (Brantley and Benveniste 2008). We focused on the Y705-phosphorylation, because it has been shown that this canonical Stat3 phosphorylation is the better readout for Stat3 activation, rather than the S737-phosphorylation, which is needed for maximal activation of Stat3 in glioma, but is dependent on initial Y705-phosphorylation (Ganguly, Fan et al. 2018). We have now added these information in the introduction section (lines 60 to 64).

ok

-          The interest of studying Iba1, MMP2, MMP9 and snail genes.

Answer: We studied Iba1 as a marker for microglia (Hambardzumyan, Gutmann et al. 2016), because 1) about one third of the cells within the tumor of our GBM-model are microglia, which is consistent with findings in the human situation (Hambardzumyan, Gutmann et al. 2016) and it is important to check whether this changes after treatment because 2) Stat3 is known to be important for microglia activation (Planas, Soriano et al. 1996, Li and Graeber 2012), which 3) could alleviate the pro-tumorigenic microglia phenotype that is usually found in GBM (Li and Graeber 2012).

As for the analysis of Mmp2, Mmp9 and Snai1: we analyzed these genes for three reasons: 1) Those genes are known to be regulated by Stat3 signaling (Luwor, Baradaran et al. 2013, Wendt, Balanis et al. 2014) and can therefore be used as a surrogate for Stat3-activation (especially in cases where a Western Blot is not possible, like for the LCM-approach). 2) All three genes are important for GBM aggressiveness, since Mmps mediate the infiltrative phenotype via modification of the ECM (Westermarck and Kahari 1999) and Snai1 is a master regulator of epithelial-mesenchymal-transition and mediates an immunosuppressive phenotype (Kudo-Saito, Shirako et al. 2009). 3) We could previously show that stable depletion of Stat3 using lentivirally delivered shRNA that all genes are depleted as well in the same cell system (Tu2449, (Priester, Copanaki et al. 2013)).

We thank the Reviewer for bringing this to our attention and have now included these further details in the introduction section.

 ok

Material and methods section

The nanoparticle characterization is out of my expertise area then should be reviewed by another expert.

The number of repetition performed by experiments is not informed.

Answer: Thank you for bringing this to our attention. We have now provided repetition numbers.

§4.2. The authors should explain why they used two different human GBM cell lines. What about their differences?

Answer: We have used two different cell lines in order the address the known heterogeneity of GBM (Friedmann-Morvinski 2014) and because we wanted to be certain that we observe generalizable effects and not cell-line specific artifacts. U87 cells are known to be particularly dependent on STAT3 (Dasgupta, Raychaudhuri et al. 2009) and are therefore a good model to study STAT3 depletion. Mz18 were chosen as second human GBM cell line, which we also previously used to analyze JAK2/STAT3-inhibition, but which expresses rather moderate levels of pSTAT3 and might therefore be not as dependent on STAT3 asU87 (Senft, Priester et al. 2011).

The § 4.6 needs to be entirely re written. It is too tufted and information are difficult to find. Nature of cells treated, time of treatment,… etc., must be specified.

A separate § concerning 1/cell transfection (which cells?), 2/Luciferase activity 3/EGFP KO and 4/proliferation assays should be produced.

Answer: Following the Reviewer’s suggestion, we have reorganized and fine-structured this part of the Materials and Methods section. More specifically, 4.9 Cell transfection, 4.10 Determination of knockdown efficacies, 4.11 Cell proliferation and viability assays, 4.12 cell cycle analysis and 4.13 Spheroid Assay now are provided as separate subsections.

In all protocols, the cell lines are now specified. Where missing, this information has also been added in the Results section. In all protocols, the time of treatment / time point of measurement or termination of the experiment is given. Additionally, we have now re-structured the Method-Section in order to provide a more coherent reading experience.

The § 4.7 needs explanations about the ways of treating the data. Indeed, the authors used the DDCt method, then the control should be fixed at 1AU, which is not the case in Fig.2b and Fig.6b. In this last fig, as the authors utilized two normalization genes (HPRT and TBP), they should be both exploited to generate a more robust result in one graph.

Answer: Thank you for pointing this out. We forgot to mention that the “+”-symbol in the Box-Whisker-Plots denotes the mean, whereas the horizontal line depicts the median. We have now added these informations into the figure legends and §4.17. Accordingly, it should now be conceivable that the control-conditions are fixed at 1.

As for Fig 2b and c: We normalized all conditions to siCtrl and not the untreated cells, since this is the correct control treatment. The Reviewer likely assumed that we normalized to untreated cells. We have now further clarified these discrepancies.

I am still annoyed by this way of treating the results because it is seemed uncorrect. Indeed, all the controls should be fixed at 1AU, then, why are there some points under and above one?

As for Fig. 6b (now 7b): In total we prepared LCM-slides for 6 samples per group. But due to the low amounts of RNA that could be isolated (and/or quality therefore) we could only obtain Ct-values for Hprt for three samples of each group. For Tbp we could obtain data from one additional sample per group. Hence, we decided to present the data independently in order to not lose the data from one animal per group. We have now clarified this in more detail in the result section.

 This point needs more explanations: if the authors can obtain 800ng RNA transcribed in cDNA, they should have sufficiently cDNA to obtain relevant Ct? It is not specified the quantity of cDNA amplified as the authors only gave the volume (4µL). I recommend launching new qRT-PCR experiments with a higher cDNA quantity. If the qRT has been performed with 2ng cDNA, the authors could use 10ng or more ng to obtain relevant Ct values and exploit correctly these data. Indeed, there are very important for the robustness of the paper, particularly Stat3 transcript expression, as the protein expression was unchanged.

The § 4.10 needs revision: how many mice were implanted, how many mice per group and above all, what about toxicity of siRNA treatment? Weights of mice should be given as well as hepatic toxicity information.

Answer: We have now added the requested information. We have shown previously that no hepatotoxicity was observed even after repeated systemic administration of our LPP over > 2 weeks (Ewe, Panchal et al. 2017). Since even this previous mode of injection, favoring hepatic delivery, did not yield toxic effects, no hepatotoxicity can be expected from local application of considerably smaller amounts (see below). We thus did not repeat hepatotoxicity experiments in this study. Nonetheless, the animals were monitored daily (sometimes twice a day) and we did not observe any detectable signs of toxicity.

The average mouse weight at the time point of the experiment was ~ 25 g. This information has been added to the manuscript.

Application of 0.5µg siRNA must be discussed: why this quantity, whereas it was used 0.8µg siRNA on 30000 cells?

Answer: Due to limitations regarding the total lipopolyplex volume we were allowed to inject, a lipopolyplex amount equivalent to 0.5 µg siRNA was the maximum possible in this experimental setting. One should also bear in mind, however, that in previous in vivo experiments we had used LPP containing only 10 µg siRNA for **systemic** injection (Ewe, Panchal et al. 2017). Given the ratio body weight / brain weight (of course, incorrectly assuming a homogenous biodistribution within the whole body after systemic injection), this means that the amounts here used for local application were even higher than in the previous in vivo studies.

Ok, in this case, why not using 0.5µg for in vitro experiments?

To make this point (limitations regarding the total lipopolyplex volume) clearer, we have made appropriate additions to the text (Materials and Methods, Discussion).

Cervical dislocation sacrifice here must be discussed as this king of method should be completely avoided to prevent brain damage at the moment of the dislocation.

Answer: It is true that, if done carelessly, cervical dislocation can lead to brain damage. However we did this on mice that received a lethal injection of an anesthetic and therefore very little force was required. Besides, even if some damage might have been afflicted this would likely be restricted to brain stem, the pons or the cerebellum, anatomic structures that are not important for this study. We regret that we had forgotten to mention the lethal injection in our first submission and have now included this information.

Even when done carefully, this method must be given up in neuroscience studies. Intra-cardiac perfusion preserves the brain tissue.

Results section

Globally, the result part is quite difficult to read as some information are missing (see M&M section) and because the resolution of all the figures is too low combined with a too small police. Moreover, the legends should give much more material to enhance the comprehension of the experiments.

Answer: We regret that the Reviewer deemed our figures of low quality/resolution. We can only assume that this was due to the processing during the upload and/or the embedding in MS Word, because we prepared all figures from high resolution and/or vector graphics and exported them as 600 dpi figures. We will of course provide the high resolution figures as separate files should this issue persist. As for the figure legends: we have now added more information and are confident that the readability is now vastly improved.

For example, in the Fig.1e, the reader has any information about the utility of Luc3 vs Luc 2, the cells used (we suppose Tu2449?).

Answer: Thank you for pointing this out. As also noted by Rev. 1 did we forget to add some information at this point. We regret this mistake and have now added the missing information.

Additionally, most information is given in the text (Materials and Methods, Results) rather than in the Figure legends. Following the Reviewer’s suggestion, however, we have now enhanced the figure legends. We revised the Figure legends and added missing information for clarification. By making appropriate additions, more information is now given in the Figure legends.

The Reviewer is correct that information on Luc2 vs. Luc3 was entirely missing and that the difference between siRNAs Luc2 and Luc3 had not been properly explained. These siRNAs target the Promega Luciferase vectors pGL2 and pGL3, respectively. In the reporter cell lines used in this study, pGL3 is stably integrated; thus, siRNA Luc3 is the specific siRNA while Luc2 served as negative control (Fig. 1g). In Fig. 1f siRNA EGFP is the specific siRNA, with again siLuc2 serving as negative control. In Fig. 1e, both siRNAs (Luc2 and Luc3) are shown simply to make the point that (absence of) toxicity is independent of the siRNA sequences, i.e., identical results are obtained for both siRNAs.

§2.2: the cell cycle results should be added in this section as they are directly linked to proliferation ones. Why the authors did not give U87 and Mz18 cell cycle data?

Answer: We have now moved the entire figure (previously Figure S3) into the main text and have labeled it as new Figure 3. In the light of the subsequent in vivo data obtained in our syngeneic mouse model, we figured that we should rather focus on the murine cell line. However, following the Reviewer’s suggestion, we have added U87 and Mz18 cell cycle data to the revised manuscript (new Suppl. Fig. S3 b, c). Please note that these data were obtained in the presence of nocodazole. Thus, we have also added a description of the procedure to the Suppl. Materials and Methods section as well as an appropriate explanation to the Figure legend.

To be noted that all the experiments, particularly cell proliferation experiments would have been more relevant on Tu and U87 spheroids, which mimics the tumor, facilitating the link with in vivo results.

Answer: We did perform cell proliferation in Tu2449 cells (Fig. 2 g and h). In fact, we even compared proliferation changes after siRNA transfection using a conventional transfection reagent (Interferin, Fig. 2g) and also tested LPP in vitro (Fig. 2h) to provide functional proof of LPP-mediated siRNA-delivery. We have now further emphasized these points in the result section and apologize for not describing it clearly enough.

Ok

As for the experiments in spheroids: We did analyze changes in sphere growth after LPP siStat3 using Tu2449 spheroids (previously Fig S3 b and C; now Fig. 3 b and c) and could successfully show growth inhibition. We only performed these experiments using Tu2449 because 1) we wanted to proof functional validity of LPP siRNA-delivery in a more complex model and 2) the Tu2449 cell line is the line used for the in vivo experiments and should therefore be analyzed in more detail compared to the human GBM cell lines. Considering that we perform siRNA-mediated Stat3-depletion we believe that these experiments are the most relevant, because the siRNAs aren’t interchangeable between human und murine cancers. Hence, by using LPP siStat3 for Tu2449 Proliferation and sphere growth we provide functional proof of 1) siRNA efficiency and 2) LPP-mediated delivery of siRNA.

OK

§ 2.3: Why the authors did not design an untreated group?

Answer: We did not design an untreated group because the applied mouse model is very well established (Pohl, Wick et al. 1999) and untreated tumor-bearing mice can hardly be compared to control-treated mice since they will not experience the same level of stress as treated mice. Since the treatment regimen is rather stressful (in total 6 intracranial injections (1 for tumor implantation; 5 for treatment); all under deep anesthesia) we reasoned that an untreated group is a very unreliable control. The nanoparticle treatment may in theory be well associated with non-specific effects from the nanoparticles, or their polymeric, lipid or RNA contents. Thus, the by far most important negative control is precisely the same treatment with the same formulation, except for using a non-specific, irrelevant siRNA. It is one beauty of the siRNA field that the design of negative controls is very straight forward by providing all components and only altering the siRNA sequence. We therefore focused on this negative control group.

 OK

§2.4/2.5: the main issue here is the stable expression of stat3, after the treatment. Normally, the gene therapy aims at decreasing this expression, then, how the authors can explain this result? It is also deceiving that Ki67 expression did not decrease.

Answer: We agree with the Reviewer that a robust decrease in Stat3 expression would have been expected, especially considering the pronounced increase in survival. We hypothesize that 1) intratumural diffusion/distribution of the LPP complexes is somewhat limited and should be addressed in due detail in future studies, maybe even in a comparative manner in different cancers. We propose that only a subset of tumor cells is reached, while the rest can retain high Stat3 expression. 2) We chose a rather aggressive GBM model that usually kills the animals after 21 days and leads to very large tumors, as previously shown (Priester, Copanaki et al. 2013). Hence, we believe that those cells that are targeted by LPPsiStat3 will have a growth disadvantage (thus, explaining the decrease in tumor size), but those will very quickly be overgrown by other cells. This in turn might also explain why we only observed a tendency towards Stat3 depletion. We regret if we had not discussed this in due detail and have now extended the discussion section concerning these discrepancies.

Another explanation could be the lifetime or the turn-over of Stat3 protein? As the injection frequencies were spaced every three days, it is possible that the cell production of Stat3 did not compensate the decrease due to siRNA delivery. Did the authors check the lifetime of this protein, and tried different frequencies of delivery of LPP siSTAT3 to validate their model?

The authors tried to answer to this problem by measuring Stat3 transcript expression, which seemed slightly decreased, despite some inconsistency about data analyses (specified above). Moreover, line 220-221, authors stipulated that the number of samples treated depended on the nature of the house keeping gene used. Please explain.

Answer: Obviously, the number of samples analyzed did not depend on the housekeeping gene and we regret if our writing was not specific enough. We performed laser-capture-microdissection (LCM) of 6 animals per group, but did not only obtain quantifiable data for 4 animals per group when measuring Tbp expression (previously Fig. 6b, now 7b) and 3 animals per group for Hprt (previously Fig. 6c, now 7c). Therefore we present the data normalized to both housekeepers separately. Of course the three samples per group normalized to Hprt could also been normalized to Tbp. We could just analyze one additional sample if normalized to Tbp. We have now added further explanations about these matters and are confident that the clarity of the chosen approach is now improved.

 As specified above, a new set of qPCR experiments with higher cDNA quantity and normalization on two or three HKG should resolve this issue. This point is very important to being the proof that siRNA injection may decrease Stat3 transcript expression.

Consequently, Fig.6 a and b are not receivable, especially since statistical analyses are not informed.

Answer: We regret that we missed to provide details about statistical analyses in the figure legends and have now added this information. As already stated in the main text was the only significant difference a reduction of Mmp9 expression if normalized to Tbp. We also stated that this is likely due to the low sample sizes. Nonetheless do we believe that the data obtained from the LCM-approach provides important insights that can explain some of the previously discussed discrepancies.

It will depend on the new set of experiment results.

Then, the conclusion about the model and the in vivo results is not acceptable, and the last paragraph before the discussion must be moderated.

Answer: We agree that our initial statement were rather strong and not fully supported by the data. We have now re-written this paragraph with more caution. The text reads as follows (lines 300 to 305):

These results support our previously proposed hypothesis that LPP siStat3 can specifically target cancer cells to limit tumor growth in vivo. Despite the robust effects of LPP-formulated siStat3 on tumor growth, our results also indicate that only a small fraction of tumor cells can be reached using the current formulations and mode of application. These observations suggest that improvement of the bioavailability may allow further enhancement of therapeutic efficacies of this approach in vivo.

If possible this last experiment should be performed again by analyzing Stat3 protein level by WB, to be sure it is less expressed into the center of the tumor.

Answer: Unfortunately, this is not possible. During the LCM-approach we used up most of the material. In fact, we initially reasoned that LCM followed by RNA-Isolation using dedicated Kits and cDNA-Synthesis using state-of-the-art Reverse Transcriptases (SS IV Vilo; life technologies) would likely offer the most output in terms of RNA quantitiy and possible transcripts. Additionally, we routinely use Taqman-probes for qRT-PCR which usually have smaller transcripts and are therefore suitable for amplification of complex, potentially degraded, samples.

 OK

Discussion section

All this section should be moderated to a better fitting with the results obtained. For example, lines 271-272.

The authors should discuss the new methods used to administer drugs in situ, such as multiport catheter.

The authors should discuss the toxicity associated with gene therapy by siRNA and the specificity of siRNA sequence.

Answer: We agree with the Reviewer that we should have been more cautious with our statements and we have carefully edited the discussion. We also extended the discussion about novel delivery methods and provide more details about the siRNA. We thank the Reviewer for these suggestions.

 The discussion is now fine.

Author Response

Reviewer 1

Dear Editor,

The authors assessed a new gene therapy efficacy by intra-cranially injected siRNA-STAT3, complexed into a lipopolyplexes (LPP), in mice implanted at the striatum level with murine glioma cells. Before the in vivo experiments, the authors in vitro evaluated the anticancer potential of the LPP-siRNA-STAT3.

This work is interesting because it addresses different main problems to treat GBM: a specific delivery, a specific target of glioma cells and avoiding degradation of the drug after the delivery.

The authors try to reach all these points, which probably constituted a promising future, but the manuscript suffers from some weaknesses, mainly about experimental approach. They are listed below.

Introduction section

It is well constructed but some information are missing about

-          Phosphorylation status of Stat3, particularly the form pY705Stat3, used in different experiments.

-          The interest of studying Iba1, MMP2, MMP9 and snail genes.

Material and methods section

The nanoparticle characterization is out of my expertise area then should be reviewed by another expert.

The number of repetition performed by experiments is not informed.

§4.2. The authors should explain why they used two different human GBM cell lines. What about their differences?

The § 4.6 needs to be entirely re written. It is too tufted and information are difficult to find. Nature of cells treated, time of treatment,… etc., must be specified.

A separate § concerning 1/cell transfection (which cells?), 2/Luciferase activity 3/EGFP KO and 4/proliferation assays should be produced.

The § 4.7 needs explanations about the ways of treating the data. Indeed, the authors used the DDCt method, then the control should be fixed at 1AU, which is not the case in Fig.2b and Fig.6b. In this last fig, as the authors utilized two normalization genes (HPRT and TBP), they should be both exploited to generate a more robust result in one graph.

The § 4.10 needs revision: how many mice were implanted, how many mice per group and above all, what about toxicity of siRNA treatment? Weights of mice should be given as well as hepatic toxicity information.

Application of 0.5µg siRNA must be discussed: why this quantity, whereas it was used 0.8µg siRNA on 30000 cells?

Cervical dislocation sacrifice here must be discussed as this king of method should be completely avoided to prevent brain damage at the moment of the dislocation.

Results section

Globally, the result part is quite difficult to read as some information are missing (see M&M section) and because the resolution of all the figures is too low combined with a too small police. Moreover, the legends should give much more material to enhance the comprehension of the experiments.

For example, in the Fig.1e, the reader has any information about the utility of Luc3 vs Luc 2, the cells used (we suppose Tu2449?).

§2.2: the cell cycle results should be added in this section as they are directly linked to proliferation ones. Why the authors did not give U87 and Mz18 cell cycle data?

To be noted that all the experiments, particularly cell proliferation experiments would have been more relevant on Tu and U87 spheroids, which mimics the tumor, facilitating the link with in vivo results.

§ 2.3: Why the authors did not design an untreated group?

§2.4/2.5: the main issue here is the stable expression of stat3, after the treatment. Normally, the gene therapy aims at decreasing this expression, then, how the authors can explain this result? It is also deceiving that Ki67 expression did not decrease.

The authors tried to answer to this problem by measuring Stat3 transcript expression, which seemed slightly decreased, despite some inconsistency about data analyses (specified above). Moreover, line 220-221, authors stipulated that the number of samples treated depended on the nature of the house keeping gene used. Please explain.

Consequently, Fig.6 a and b are not receivable, especially since statistical analyses are not informed. Then, the conclusion about the model and the in vivo results is not acceptable, and the last paragraph before the discussion must be moderated.

If possible this last experiment should be performed again by analyzing Stat3 protein level by WB, to be sure it is less expressed into the center of the tumor.

Discussion section

All this section should be moderated to a better fitting with the results obtained. For example, lines 271-272.

The authors should discuss the new methods used to administer drugs in situ, such as multiport catheter.

The authors should discuss the toxicity associated with gene therapy by siRNA and the specificity of siRNA sequence.

Second round

Dear Editor,

The authors assessed a new gene therapy efficacy by intra-cranially injected siRNA-STAT3, complexed into a lipopolyplexes (LPP), in mice implanted at the striatum level with murine glioma cells. Before the in vivo experiments, the authors in vitro evaluated the anticancer potential of the LPP-siRNA-STAT3.

This work is interesting because it addresses different main problems to treat GBM: a specific delivery, a specific target of glioma cells and avoiding degradation of the drug after the delivery.

The authors try to reach all these points, which probably constituted a promising future, but the manuscript suffers from some weaknesses, mainly about experimental approach. They are listed below.

Answer: We appreciate the Reviewer’s overall positive assessment of our work. His / her specific points are addressed in the revised manuscript as detailed below:

Introduction section

It is well constructed but some information are missing about

-          Phosphorylation status of Stat3, particularly the form pY705Stat3, used in different experiments.

Answer: We assume the Reviewer asks for clarification about the relevance of pStat3 and the reason why we specifically analyzed pY705Stat3. To regulate gene expression, Stat3 needs to be phosphorylated by many upstream kinases including Janus Kinases (JAK2) (Avalle, Camporeale et al. 2017). Hence pStat3 is a good marker for active Stat3-signaling (Brantley and Benveniste 2008). We focused on the Y705-phosphorylation, because it has been shown that this canonical Stat3 phosphorylation is the better readout for Stat3 activation, rather than the S737-phosphorylation, which is needed for maximal activation of Stat3 in glioma, but is dependent on initial Y705-phosphorylation (Ganguly, Fan et al. 2018). We have now added these information in the introduction section (lines 60 to 64).

ok

-          The interest of studying Iba1, MMP2, MMP9 and snail genes.

Answer: We studied Iba1 as a marker for microglia (Hambardzumyan, Gutmann et al. 2016), because 1) about one third of the cells within the tumor of our GBM-model are microglia, which is consistent with findings in the human situation (Hambardzumyan, Gutmann et al. 2016) and it is important to check whether this changes after treatment because 2) Stat3 is known to be important for microglia activation (Planas, Soriano et al. 1996, Li and Graeber 2012), which 3) could alleviate the pro-tumorigenic microglia phenotype that is usually found in GBM (Li and Graeber 2012).

As for the analysis of Mmp2, Mmp9 and Snai1: we analyzed these genes for three reasons: 1) Those genes are known to be regulated by Stat3 signaling (Luwor, Baradaran et al. 2013, Wendt, Balanis et al. 2014) and can therefore be used as a surrogate for Stat3-activation (especially in cases where a Western Blot is not possible, like for the LCM-approach). 2) All three genes are important for GBM aggressiveness, since Mmps mediate the infiltrative phenotype via modification of the ECM (Westermarck and Kahari 1999) and Snai1 is a master regulator of epithelial-mesenchymal-transition and mediates an immunosuppressive phenotype (Kudo-Saito, Shirako et al. 2009). 3) We could previously show that stable depletion of Stat3 using lentivirally delivered shRNA that all genes are depleted as well in the same cell system (Tu2449, (Priester, Copanaki et al. 2013)).

We thank the Reviewer for bringing this to our attention and have now included these further details in the introduction section.

 ok

Material and methods section

The nanoparticle characterization is out of my expertise area then should be reviewed by another expert.

The number of repetition performed by experiments is not informed.

Answer: Thank you for bringing this to our attention. We have now provided repetition numbers.

§4.2. The authors should explain why they used two different human GBM cell lines. What about their differences?

Answer: We have used two different cell lines in order the address the known heterogeneity of GBM (Friedmann-Morvinski 2014) and because we wanted to be certain that we observe generalizable effects and not cell-line specific artifacts. U87 cells are known to be particularly dependent on STAT3 (Dasgupta, Raychaudhuri et al. 2009) and are therefore a good model to study STAT3 depletion. Mz18 were chosen as second human GBM cell line, which we also previously used to analyze JAK2/STAT3-inhibition, but which expresses rather moderate levels of pSTAT3 and might therefore be not as dependent on STAT3 asU87 (Senft, Priester et al. 2011).

The § 4.6 needs to be entirely re written. It is too tufted and information are difficult to find. Nature of cells treated, time of treatment,… etc., must be specified.

A separate § concerning 1/cell transfection (which cells?), 2/Luciferase activity 3/EGFP KO and 4/proliferation assays should be produced.

Answer: Following the Reviewer’s suggestion, we have reorganized and fine-structured this part of the Materials and Methods section. More specifically, 4.9 Cell transfection, 4.10 Determination of knockdown efficacies, 4.11 Cell proliferation and viability assays, 4.12 cell cycle analysis and 4.13 Spheroid Assay now are provided as separate subsections.

In all protocols, the cell lines are now specified. Where missing, this information has also been added in the Results section. In all protocols, the time of treatment / time point of measurement or termination of the experiment is given. Additionally, we have now re-structured the Method-Section in order to provide a more coherent reading experience.

The § 4.7 needs explanations about the ways of treating the data. Indeed, the authors used the DDCt method, then the control should be fixed at 1AU, which is not the case in Fig.2b and Fig.6b. In this last fig, as the authors utilized two normalization genes (HPRT and TBP), they should be both exploited to generate a more robust result in one graph.

Answer: Thank you for pointing this out. We forgot to mention that the “+”-symbol in the Box-Whisker-Plots denotes the mean, whereas the horizontal line depicts the median. We have now added these informations into the figure legends and §4.17. Accordingly, it should now be conceivable that the control-conditions are fixed at 1.

As for Fig 2b and c: We normalized all conditions to siCtrl and not the untreated cells, since this is the correct control treatment. The Reviewer likely assumed that we normalized to untreated cells. We have now further clarified these discrepancies.

I am still annoyed by this way of treating the results because it is seemed uncorrect. Indeed, all the controls should be fixed at 1AU, then, why are there some points under and above one?

Answer 2: We are very sorry, but not all control values should be fixed at 1 a.u., but rather the mean of the entire group should be (which is the case in the Figure), because if all individual values of this group would be fixed at 1, its data distribution would be lost.

For clarification: First we calculate dCt by subtracting CTtarget gene from CThousekeeper per sample. Than we subtract dCTsample1 from dCTmean_of_all_samples_of_control-group. This way the mean expression of the control group is 1 a.u. (depicted by the “+”-Symbol in the box-whisker-plots), but we still retain the data distribution of the individual samples in this group, which we feel contributes to the completeness of the data shown. There are numerous online tutorials (e.g.: https://toptipbio.com/delta-delta-ct-pcr/) and publications (Livak and Schmittgen 2001) showing exactly this way of treating similar data.

As for Fig. 6b (now 7b): In total we prepared LCM-slides for 6 samples per group. But due to the low amounts of RNA that could be isolated (and/or quality therefore) we could only obtain Ct-values for Hprt for three samples of each group. For Tbp we could obtain data from one additional sample per group. Hence, we decided to present the data independently in order to not lose the data from one animal per group. We have now clarified this in more detail in the result section.

 This point needs more explanations: if the authors can obtain 800ng RNA transcribed in cDNA, they should have sufficiently cDNA to obtain relevant Ct? It is not specified the quantity of cDNA amplified as the authors only gave the volume (4µL). I recommend launching new qRT-PCR experiments with a higher cDNA quantity. If the qRT has been performed with 2ng cDNA, the authors could use 10ng or more ng to obtain relevant Ct values and exploit correctly these data. Indeed, there are very important for the robustness of the paper, particularly Stat3 transcript expression, as the protein expression was unchanged.

Answer2: We would like to thank the reviewer for pointing this out. This is a misunderstanding caused by insufficent information given by us for the respective method in the Materials and Methods section. The RNA samples obtained from the LCM approach were handled differently than those from the in vitro experiments. We performed the RNA isolation from microdissected FFPE tissue according to the Arcturus FFPE RNA Isolation Kit and we eluted the RNA in 12 µl. From our pre-tests of the LCM method, we already knew that obtained RNA contents are very low (for one to two pooled sections between 1.9 and 3.6 ng/µl; ~25-45 ng in total), so we decided not to waste any RNA for quantification, but to transcribe the whole samples into cDNA using SuperScript Vilo IV. Based on the recommendations given in the RNA Isolation Kit used, we immediately had transcribed the RNA into cDNA and performed the initial qPCR within 2-3 days. Following the reviewer´s recommendation, we also tried to repeat the qPCR experiment with the remaining cDNA. Unfortunately, the technical quality of this experiment was insufficient, most likely due to sample degradation in the cDNA over time.

To improve the Figure, we have now re-analyzed the existing data with normalization to both housekeepers as suggested by the reviewer (new Fig. 7b), obtaining similar data compared to the old Figure. We also increased the width of the figure, thereby further improving its comprehensibility and clarity. We would like to apologize for the missing information given in the former version of the manuscript and have now also added the respective information in the Materials and Methods section.

The § 4.10 needs revision: how many mice were implanted, how many mice per group and above all, what about toxicity of siRNA treatment? Weights of mice should be given as well as hepatic toxicity information.

Answer: We have now added the requested information. We have shown previously that no hepatotoxicity was observed even after repeated systemic administration of our LPP over > 2 weeks (Ewe, Panchal et al. 2017). Since even this previous mode of injection, favoring hepatic delivery, did not yield toxic effects, no hepatotoxicity can be expected from local application of considerably smaller amounts (see below). We thus did not repeat hepatotoxicity experiments in this study. Nonetheless, the animals were monitored daily (sometimes twice a day) and we did not observe any detectable signs of toxicity.

The average mouse weight at the time point of the experiment was ~ 25 g. This information has been added to the manuscript.

Application of 0.5µg siRNA must be discussed: why this quantity, whereas it was used 0.8µg siRNA on 30000 cells?

Answer: Due to limitations regarding the total lipopolyplex volume we were allowed to inject, a lipopolyplex amount equivalent to 0.5 µg siRNA was the maximum possible in this experimental setting. One should also bear in mind, however, that in previous in vivo experiments we had used LPP containing only 10 µg siRNA for **systemic** injection (Ewe, Panchal et al. 2017). Given the ratio body weight / brain weight (of course, incorrectly assuming a homogenous biodistribution within the whole body after systemic injection), this means that the amounts here used for local application were even higher than in the previous in vivo studies.

Ok, in this case, why not using 0.5µg for in vitro experiments?

Answer2: In many respects, the in vitro situation in a well format is not comparable to the in vivo conditions upon direct injection into the tissue. In the well, the complexes are ideally mixed and distributed throughout the whole volume while in vivo this cannot be expected due to diffusion being restricted by the tissue stroma and other barriers. On the other hand, this may well lead to longer interaction times of the complexes with the cells in vivo, while in the well the diffusion is faster, also leading to more rapid desorption processes from the cells’ surfaces. Since the extracellular matrix of the cells may be different in vitro vs. in vivo as well, uptake mechanisms may vary as well. For these reasons, we have often observed that in vitro results can only partially be extrapolated onto the in vivo situation, i.e., that best performing nanoparticles in vitro are not necessarily the candidates of choice in the in vivo situation. While this, first of all, highlights the necessity of performing in vivo experiments, it also emphasizes that using exactly the same amounts in two (in vitro – in vivo) given experimental settings may not be a major requirement.

The amount of 0.8 µg siRNA per well has been long established in our lab as one standard format, which nicely allows to compare different nanoparticles by providing sufficient amounts for specific knockdown effects while avoiding extensive non-specific effects.

Still following the Reviewer’s thought, however, one could go even one step further. The mass of the mouse brain is ~ 0.5 g or ~ 0.45 ml while the total volume per well of a 24-well plate was 1 ml. So, using the larger siRNA amount (0.8 µg siRNA) brings us even closer to the in vivo situation.

To make this point (limitations regarding the total lipopolyplex volume) clearer, we have made appropriate additions to the text (Materials and Methods, Discussion).

Cervical dislocation sacrifice here must be discussed as this king of method should be completely avoided to prevent brain damage at the moment of the dislocation.

Answer: It is true that, if done carelessly, cervical dislocation can lead to brain damage. However we did this on mice that received a lethal injection of an anesthetic and therefore very little force was required. Besides, even if some damage might have been afflicted this would likely be restricted to brain stem, the pons or the cerebellum, anatomic structures that are not important for this study. We regret that we had forgotten to mention the lethal injection in our first submission and have now included this information.

Even when done carefully, this method must be given up in neuroscience studies. Intra-cardiac perfusion preserves the brain tissue.

Answer2: Thank you for this suggestion. We agree that perfusion preserves the tissue much better and we will consider this method in future studies, since it also preserves the entire mouse for additional analyses. In all honesty, cervical dislocation of deeply/deadly anesthetized animals was also chosen for practical reasons, since the experiments were performed by only one experimentator and perfusion is very time-consuming. Nonetheless, we believe that the approach used here did not adversely affect the main message of our manuscript, as we did not observe any tissue damage in our histological examinations.

Results section

Globally, the result part is quite difficult to read as some information are missing (see M&M section) and because the resolution of all the figures is too low combined with a too small police. Moreover, the legends should give much more material to enhance the comprehension of the experiments.

Answer: We regret that the Reviewer deemed our figures of low quality/resolution. We can only assume that this was due to the processing during the upload and/or the embedding in MS Word, because we prepared all figures from high resolution and/or vector graphics and exported them as 600 dpi figures. We will of course provide the high resolution figures as separate files should this issue persist. As for the figure legends: we have now added more information and are confident that the readability is now vastly improved.

For example, in the Fig.1e, the reader has any information about the utility of Luc3 vs Luc 2, the cells used (we suppose Tu2449?).

Answer: Thank you for pointing this out. As also noted by Rev. 1 did we forget to add some information at this point. We regret this mistake and have now added the missing information.

Additionally, most information is given in the text (Materials and Methods, Results) rather than in the Figure legends. Following the Reviewer’s suggestion, however, we have now enhanced the figure legends. We revised the Figure legends and added missing information for clarification. By making appropriate additions, more information is now given in the Figure legends.

The Reviewer is correct that information on Luc2 vs. Luc3 was entirely missing and that the difference between siRNAs Luc2 and Luc3 had not been properly explained. These siRNAs target the Promega Luciferase vectors pGL2 and pGL3, respectively. In the reporter cell lines used in this study, pGL3 is stably integrated; thus, siRNA Luc3 is the specific siRNA while Luc2 served as negative control (Fig. 1g). In Fig. 1f siRNA EGFP is the specific siRNA, with again siLuc2 serving as negative control. In Fig. 1e, both siRNAs (Luc2 and Luc3) are shown simply to make the point that (absence of) toxicity is independent of the siRNA sequences, i.e., identical results are obtained for both siRNAs.

§2.2: the cell cycle results should be added in this section as they are directly linked to proliferation ones. Why the authors did not give U87 and Mz18 cell cycle data?

Answer: We have now moved the entire figure (previously Figure S3) into the main text and have labeled it as new Figure 3. In the light of the subsequent in vivo data obtained in our syngeneic mouse model, we figured that we should rather focus on the murine cell line. However, following the Reviewer’s suggestion, we have added U87 and Mz18 cell cycle data to the revised manuscript (new Suppl. Fig. S3 b, c). Please note that these data were obtained in the presence of nocodazole. Thus, we have also added a description of the procedure to the Suppl. Materials and Methods section as well as an appropriate explanation to the Figure legend.

To be noted that all the experiments, particularly cell proliferation experiments would have been more relevant on Tu and U87 spheroids, which mimics the tumor, facilitating the link with in vivo results.

Answer: We did perform cell proliferation in Tu2449 cells (Fig. 2 g and h). In fact, we even compared proliferation changes after siRNA transfection using a conventional transfection reagent (Interferin, Fig. 2g) and also tested LPP in vitro (Fig. 2h) to provide functional proof of LPP-mediated siRNA-delivery. We have now further emphasized these points in the result section and apologize for not describing it clearly enough.

Ok

As for the experiments in spheroids: We did analyze changes in sphere growth after LPP siStat3 using Tu2449 spheroids (previously Fig S3 b and C; now Fig. 3 b and c) and could successfully show growth inhibition. We only performed these experiments using Tu2449 because 1) we wanted to proof functional validity of LPP siRNA-delivery in a more complex model and 2) the Tu2449 cell line is the line used for the in vivo experiments and should therefore be analyzed in more detail compared to the human GBM cell lines. Considering that we perform siRNA-mediated Stat3-depletion we believe that these experiments are the most relevant, because the siRNAs aren’t interchangeable between human und murine cancers. Hence, by using LPP siStat3 for Tu2449 Proliferation and sphere growth we provide functional proof of 1) siRNA efficiency and 2) LPP-mediated delivery of siRNA.

OK

§ 2.3: Why the authors did not design an untreated group?

Answer: We did not design an untreated group because the applied mouse model is very well established (Pohl, Wick et al. 1999) and untreated tumor-bearing mice can hardly be compared to control-treated mice since they will not experience the same level of stress as treated mice. Since the treatment regimen is rather stressful (in total 6 intracranial injections (1 for tumor implantation; 5 for treatment); all under deep anesthesia) we reasoned that an untreated group is a very unreliable control. The nanoparticle treatment may in theory be well associated with non-specific effects from the nanoparticles, or their polymeric, lipid or RNA contents. Thus, the by far most important negative control is precisely the same treatment with the same formulation, except for using a non-specific, irrelevant siRNA. It is one beauty of the siRNA field that the design of negative controls is very straight forward by providing all components and only altering the siRNA sequence. We therefore focused on this negative control group.

 OK

§2.4/2.5: the main issue here is the stable expression of stat3, after the treatment. Normally, the gene therapy aims at decreasing this expression, then, how the authors can explain this result? It is also deceiving that Ki67 expression did not decrease.

Answer: We agree with the Reviewer that a robust decrease in Stat3 expression would have been expected, especially considering the pronounced increase in survival. We hypothesize that 1) intratumural diffusion/distribution of the LPP complexes is somewhat limited and should be addressed in due detail in future studies, maybe even in a comparative manner in different cancers. We propose that only a subset of tumor cells is reached, while the rest can retain high Stat3 expression. 2) We chose a rather aggressive GBM model that usually kills the animals after 21 days and leads to very large tumors, as previously shown (Priester, Copanaki et al. 2013). Hence, we believe that those cells that are targeted by LPPsiStat3 will have a growth disadvantage (thus, explaining the decrease in tumor size), but those will very quickly be overgrown by other cells. This in turn might also explain why we only observed a tendency towards Stat3 depletion. We regret if we had not discussed this in due detail and have now extended the discussion section concerning these discrepancies.

Another explanation could be the lifetime or the turn-over of Stat3 protein? As the injection frequencies were spaced every three days, it is possible that the cell production of Stat3 did not compensate the decrease due to siRNA delivery. Did the authors check the lifetime of this protein, and tried different frequencies of delivery of LPP siSTAT3 to validate their model?

Answer2: We would like to thank the reviewer for this excellent remark. Unfortunately we did not test different treatment frequencies for the treatment, because we based the treatment schedule on in vitro data showing depletion of 72h and more pronounced after 96h of LPP siRNA treatment. Hence we reasoned that a treatment every third day will result in robust depletion of Stat3 over time. Obviously, there are remarkable differences between the in vitro and in vivo situation and we have now further discussed these issues in the discussion section (lines 376 to 384; clean version).

A different explanation for the low Stat3-depletion could be changes in Stat3 turnover or stability. Accordingly, the half-life of Stat3 has been previously determined to be between 30h in primary murine neurons to 90h in hepatocytes [53] and around 50h in Epstein-Barr-Virus-infected PBMCs [54] indicating pronounced variations between cell types. Considering that HeLa cells have a 50% turnover rate of ~60% of their entire proteome within 5h [55] it might also be possible that the in vivo microenvironment stimulates the tumor cells promoting faster protein turnover, because of increased cell divisions. In this case the treatment frequency of every third might be too low. However, the fact that we already observe strong increases in survival by 7 days (~25%) indicates that by continuous delivery of siRNA this issue can likely be resolved.

It should also be noted that, over the whole period of a therapy study, the frequencies of the LPP delivery may not be a critical parameter. Several steps require some time or are likely to prevail for some time: the diffusion of the LPP to the target cells will certainly not be a rapid process, leading to delays in cellular uptake. The intracellular release of the siRNA from the LPP formulations is based on a slow nanoparticle decomposition rather than a burst release as to be expected from pure liposomal formulations. Thus, the intracellular availability of siRNAs and the onset of the knockdown will gradually kick in over a wider time period. On the other hand, siRNAs have been shown, once incorporated into RISC, to be stable for a longer time period. In fact, in cell culture the decrease of knockdown efficacy is rather determined by cell doubling times, with fast cell cycle progression and cell proliferation leading to the intracellular “dilution” of the siRNAs due to their distribution into the daughter cells. Even in the case of the proliferation-active tumor xenograft tissue, the doubling times are considerably lower (resembling more closely the in vivo situation in a patient) and thus siRNA activities can be expected to last over a longer time period of at least several days.

While final proof would require a microscopic evaluation on the single cell level over time, which is not easily feasible and beyond the scope of this paper, we stick to our hypothesis that the LPP distribution, i.e., the extent of nanoparticles reaching the various target cells, poses a major limitation.

This is also supported by previous experiments from our group on nanoparticle tissue penetration – see below (Reviewers 2 and 3).

The authors tried to answer to this problem by measuring Stat3 transcript expression, which seemed slightly decreased, despite some inconsistency about data analyses (specified above). Moreover, line 220-221, authors stipulated that the number of samples treated depended on the nature of the house keeping gene used. Please explain.

Answer: Obviously, the number of samples analyzed did not depend on the housekeeping gene and we regret if our writing was not specific enough. We performed laser-capture-microdissection (LCM) of 6 animals per group, but did not only obtain quantifiable data for 4 animals per group when measuring Tbp expression (previously Fig. 6b, now 7b) and 3 animals per group for Hprt (previously Fig. 6c, now 7c). Therefore we present the data normalized to both housekeepers separately. Of course the three samples per group normalized to Hprt could also been normalized to Tbp. We could just analyze one additional sample if normalized to Tbp. We have now added further explanations about these matters and are confident that the clarity of the chosen approach is now improved.

 As specified above, a new set of qPCR experiments with higher cDNA quantity and normalization on two or three HKG should resolve this issue. This point is very important to being the proof that siRNA injection may decrease Stat3 transcript expression.

Answer2: Again, we are very sorry for causing these misunderstandings. Of course using higher cDNA concentrations likely leads to more robust Ct-values. As explained above, we tried to repeat the qPCR experiment with the remaining samples and used higher amounts of cDNA this time, but the repetition experiment was not successful. However, we have now re-analyzed the existing data with normalization to both housekeepers as suggested by the reviewer and we obtained very similar data compared to our initial analysis.

Consequently, Fig.6 a and b are not receivable, especially since statistical analyses are not informed.

Answer: We regret that we missed to provide details about statistical analyses in the figure legends and have now added this information. As already stated in the main text was the only significant difference a reduction of Mmp9 expression if normalized to Tbp. We also stated that this is likely due to the low sample sizes. Nonetheless do we believe that the data obtained from the LCM-approach provides important insights that can explain some of the previously discussed discrepancies.

It will depend on the new set of experiment results.

Answer2: As outlined above, it unfortunately was not possible to perform a new set of experiments. Even the repetition of the entire LCM-approach is unlikely to lead to any improvement since our FFPE tissue is mostly used up now. We hope that the reviewer finds our improved analysis and additional efforts now acceptable.

Then, the conclusion about the model and the in vivo results is not acceptable, and the last paragraph before the discussion must be moderated.

Answer: We agree that our initial statement were rather strong and not fully supported by the data. We have now re-written this paragraph with more caution. The text reads as follows (lines 300 to 305):

These results support our previously proposed hypothesis that LPP siStat3 can specifically target cancer cells to limit tumor growth in vivo. Despite the robust effects of LPP-formulated siStat3 on tumor growth, our results also indicate that only a small fraction of tumor cells can be reached using the current formulations and mode of application. These observations suggest that improvement of the bioavailability may allow further enhancement of therapeutic efficacies of this approach in vivo.

If possible this last experiment should be performed again by analyzing Stat3 protein level by WB, to be sure it is less expressed into the center of the tumor.

Answer: Unfortunately, this is not possible. During the LCM-approach we used up most of the material. In fact, we initially reasoned that LCM followed by RNA-Isolation using dedicated Kits and cDNA-Synthesis using state-of-the-art Reverse Transcriptases (SS IV Vilo; life technologies) would likely offer the most output in terms of RNA quantitiy and possible transcripts. Additionally, we routinely use Taqman-probes for qRT-PCR which usually have smaller transcripts and are therefore suitable for amplification of complex, potentially degraded, samples.

 OK

Discussion section

All this section should be moderated to a better fitting with the results obtained. For example, lines 271-272.

The authors should discuss the new methods used to administer drugs in situ, such as multiport catheter.

The authors should discuss the toxicity associated with gene therapy by siRNA and the specificity of siRNA sequence.

Answer: We agree with the Reviewer that we should have been more cautious with our statements and we have carefully edited the discussion. We also extended the discussion about novel delivery methods and provide more details about the siRNA. We thank the Reviewer for these suggestions.

 The discussion is now fine.

Answer2: Again we would like to thank the reviewer for his/hers helpful suggestions and we rightfully acknowledge that the manuscript and especially the discussion section is now vastly improved.

References Revision 1

Avalle, L., A. Camporeale, A. Camperi and V. Poli (2017). "STAT3 in cancer: A double edged sword." Cytokine 98: 42-50.

Brantley, E. C. and E. N. Benveniste (2008). "Signal transducer and activator of transcription-3: a molecular hub for signaling pathways in gliomas." Molecular cancer research : MCR 6(5): 675-684.

Dasgupta, A., B. Raychaudhuri, T. Haqqi, R. Prayson, E. G. Van Meir, M. Vogelbaum and S. J. Haque (2009). "Stat3 activation is required for the growth of U87 cell-derived tumours in mice." Eur J Cancer 45(4): 677-684.

Ewe, A., O. Panchal, S. R. Pinnapireddy, U. Bakowsky, S. Przybylski, A. Temme and A. Aigner (2017). "Liposome-polyethylenimine complexes (DPPC-PEI lipopolyplexes) for therapeutic siRNA delivery in vivo." Nanomedicine 13(1): 209-218.

Friedmann-Morvinski, D. (2014). "Glioblastoma heterogeneity and cancer cell plasticity." Crit Rev Oncog 19(5): 327-336.

Ganguly, D., M. Fan, C. H. Yang, B. Zbytek, D. Finkelstein, M. F. Roussel and L. M. Pfeffer (2018). "The critical role that STAT3 plays in glioma-initiating cells: STAT3 addiction in glioma." Oncotarget 9(31): 22095-22112.

Hambardzumyan, D., D. H. Gutmann and H. Kettenmann (2016). "The role of microglia and macrophages in glioma maintenance and progression." Nat Neurosci 19(1): 20-27.

Kudo-Saito, C., H. Shirako, T. Takeuchi and Y. Kawakami (2009). "Cancer metastasis is accelerated through immunosuppression during Snail-induced EMT of cancer cells." Cancer Cell 15(3): 195-206.

Li, W. and M. B. Graeber (2012). "The molecular profile of microglia under the influence of glioma." Neuro Oncol 14(8): 958-978.

Luwor, R. B., B. Baradaran, L. E. Taylor, J. Iaria, T. V. Nheu, N. Amiry, C. M. Hovens, B. Wang, A. H. Kaye and H. J. Zhu (2013). "Targeting Stat3 and Smad7 to restore TGF-beta cytostatic regulation of tumor cells in vitro and in vivo." Oncogene 32(19): 2433-2441.

Planas, A. M., M. A. Soriano, M. Berruezo, C. Justicia, A. Estrada, S. Pitarch and I. Ferrer (1996). "Induction of Stat3, a signal transducer and transcription factor, in reactive microglia following transient focal cerebral ischaemia." Eur J Neurosci 8(12): 2612-2618.

Pohl, U., W. Wick, J. Weissenberger, J. P. Steinbach, J. Dichgans, A. Aguzzi and M. Weller (1999). "Characterization of Tu-2449, a glioma cell line derived from a spontaneous tumor in GFAP-v-src-transgenic mice: comparison with established murine glioma cell lines." Int J Oncol 15(4): 829-834.

Priester, M., E. Copanaki, V. Vafaizadeh, S. Hensel, C. Bernreuther, M. Glatzel, V. Seifert, B. Groner, D. Kogel and J. Weissenberger (2013). "STAT3 silencing inhibits glioma single cell infiltration and tumor growth." Neuro-oncology 15(7): 840-852.

Senft, C., M. Priester, M. Polacin, K. Schroder, V. Seifert, D. Kogel and J. Weissenberger (2011). "Inhibition of the JAK-2/STAT3 signaling pathway impedes the migratory and invasive potential of human glioblastoma cells." Journal of neuro-oncology 101(3): 393-403.

Wendt, M. K., N. Balanis, C. R. Carlin and W. P. Schiemann (2014). "STAT3 and epithelial-mesenchymal transitions in carcinomas." JAKSTAT 3(1): e28975.

Westermarck, J. and V. M. Kahari (1999). "Regulation of matrix metalloproteinase expression in tumor invasion." FASEB J 13(8): 781-792.

References Revision2:

Livak, K. J. and T. D. Schmittgen (2001). "Analysis of relative gene expression data using real-time quantitative PCR and the 2(-Delta Delta C(T)) Method." Methods 25(4): 402-408.